# Evolution of the Chinese guarantee network under financial crisis and stimulus program

Yingli Wang[1,2,4], Qingpeng Zhang ● [3,4✉] & Xiaoguang Yang ● [1,2,4✉]

Our knowledge about the evolution of guarantee network in downturn period is limited due to the lack of comprehensive data of the whole credit system. Here we analyze the dynamic Chinese guarantee network constructed from a comprehensive bank loan dataset that accounts for nearly 80% total loans in China, during 01/2007-03/2012. The results show that, first, during the 2007-2008 global financial crisis, the guarantee network became smaller, less connected and more stable because of many bankruptcies; second, the stimulus program encouraged mutual guarantee behaviors, resulting in highly reciprocal and fragile network structure; third, the following monetary policy adjustment enhanced the resilience of the guarantee network by reducing mutual guarantees. Interestingly, our work reveals that the financial crisis made the network more resilient, and conversely, the government bailout degenerated network resilience. These counterintuitive findings can provide new insight into the resilience of real-world credit system under external shocks or rescues.

[1] The Academy of Mathematics and Systems Science, Chinese Academy of Sciences, Beijing 100190, China. [2] The University of Chinese Academy of Sciences, Beijing 100049, China. [3] School of Data Science, City University of Hong Kong, Kowloon, 00001 Hong Kong, China. [4]These authors contributed equally: Yingli Wang, Qingpeng Zhang, Xiaoguang Yang. ✉email: qingpeng.zhang@cityu.edu.hk; xgyang@iss.ac.cn

A financial network represents a collection of entities (e.g., firms and banks), linked by mutually beneficial business relationships[1,2]. The guarantee relationship between two firms represents the responsibility of a firm (guarantor) to assume the debt obligation of another firm (borrower) if the borrower failed to meet its legal obligation of a loan (default). Based on the credit networks formed by the guarantee relationship, we can understand the systemic risk better by investigating the evolution of the credit system in downturn period. In particular, it is critical to identify the risk of cascading failure in the guarantee network caused by the failure of one or a few entities, because such cascading failure could potentially disrupt the connectivity and reliability of the whole credit system[3,4]. Since the guarantee relationship can be naturally represented as networks, applying network science methods to the analysis of the cascading failures in financial networks has emerged as one of the most compelling and active research areas in economics and finance[5].

China has the biggest bank loan in the world. The credit market is the core of China's financial system, and the corresponding guarantee network might arguably be the largest credit networks in the world. With the outbreak of 2007–2008 global financial crisis, Chinese government implemented the stimulus program worth of RMB ¥4 trillion in 9 November 2008 to reduce the impact of the financial crisis on China's economy by investing hugely in infrastructure and social welfare by 2010. The credit condition was also loosen to encourage loan applications from firms[6]. This stimulus program, though successfully sustained China's economic growth and largely stabilized the world economy, has resulted in a surge in debt in China, and dramatic structural change of the credit system. There were a huge amount of loans going to state-owned enterprises (SOEs), many of which could not meet the credit standard to obtain loans before the stimulus program[7]. These SOEs could meet the loosened credit standard through guaranteeing each other[8]. Although many of these SOEs were saved by loans, economic studies suspected that the stimulus program could cause explosion of credit debt and trigger future disruptions in the financial system because of huge loans to low-quality firms[9–11].

Decision makers also recognized such risk and took actions to regulate the debt behavior. In March 2010, Chinese government announced the plan to improve macro-control regulations. Since then, the People's Bank of China (PBoC, the central bank of China) increased the reserve requirement ratio for five times in 2010 (from 15.5% to 18%). The major change of the monetary policy operations occurred on 20 October 2010, when PBoC raised the interest rates for the first time since the financial crisis. These regulations and policy adjustment, to some extent, helped to reduce the impact of the negative consequence of the stimulus program.

Despite the qualitative critics and studies on the stimulus program, little is known about the changes of the guarantee networks and how these changes are linked to the risk of cascading failures. A good understanding of the guarantee network's dynamic structure under pressure could enhance the economic policy-making through identifying the potential systemic risks such as the domino-like cascading failures. Facing the onset of financial crisis, it is of particular importance to examine the tradeoff between saving firms with government bailout and maintaining a good resilience of the financial system. There is a critical need for data-driven quantitative studies of the guarantee network and its associated risk. However, how to leverage real-world guarantee data to model such behaviors and the overall structure of the guarantee network remains a challenge.

Network science presents a natural way to address the challenge in modeling guarantee networks. Network science aims to discover the underlying patterns of interactions among elements in complex systems. It has been widely applied to modeling structure and dynamics of real-world complex systems[12]. Recently, network science has been applied to finance research. Specifically, network science has been applied extensively to analyzing the global banking system[13], international financial network[14], and interlocking boards of directors[15–17]. Please refer to the Related Work section in the Supplementary Note 1 for a detailed review.

The guarantee interdependencies can be naturally represented as networks, in which each node represents a firm, and each (directed) edge represents the guarantee relationship between the two corresponding firms. From such a guarantee network, we can capture the contagion path of obligations and failures. Using methods in network science to analyze guarantee networks could fill the research gap in data-driven insights into how the topological structure of guarantee network is associated with economic policies and contagion risks, and help decision makers identify the potential systemic risks caused by firms' failures[18].

Existing research on guarantee networks mainly focused on the analytics of small sampled data with only dozens or hundreds of firms. Although small-scale network analysis may lead to useful insights into the risk connection of individual client, it cannot tell big stories about the stability of the whole credit market, not to mention the evolution of credit networks[19–22]. Large-scale empirical studies of the nationwide guarantee networks are needed to understand the global topological properties of the guarantee system.

By harnessing a comprehensive data provided by one of China's major regulatory bodies ranging 01/2007–03/2012, we investigate the structure and evolution of Chinese guarantee network to answer the following questions: What are the unique topological properties of the Chinese guarantee network? What is the influence of 2007–2008 global financial crisis, China's stimulus program and the following monetary policy adjustment on the topological structure of Chinese guarantee network? Does the change of topological structure of Chinese guarantee network influence the resilience of the system? To the best of our knowledge, this is the first attempt to quantitatively characterize the evolution of the nationwide guarantee network. The empirical and simulation studies indicate that the financial crisis made the network more resilient, and conversely, the government bailout degenerated network resilience.

## Results

**Static topological and financial properties.** In this section, we characterize the static topological and financial properties of the guarantee network. Our dataset covers three important financial events: bankruptcy of New Century Financial Corporation in April 2007, bankruptcy of Lehman Brothers in September 2008, and implementation of the Chinese economic stimulus program from December 2008 to December 2010. According to these extreme events, we divided the data range into three phases. Phase 1 (04/2007–11/2008) is the period of global financial crisis. Bankruptcy of New Century Financial Corp. on April 2007 could be considered as the beginning of subprime mortgage crisis. Lehman Brothers Holdings Inc. filed for bankruptcy in September 2008 and global financial crisis reached the summit. Phase 2 (12/2008–12/2010) is the period when China implemented the four-trillion stimulus program. Phase 3 (01/2011–03/2012) is the adjustment period after the stimulus program.

We analyze the guarantee network in each month, and summarize the common properties for the whole period, as well as the three phases. The topological properties of the guarantee network are presented in Supplementary Table 1. Despite the

increasing size of the guarantee network, there are a set of common topological properties throughout the whole period.

First, the average in-/out-degree ($<d>$) is slightly lower than one, indicating that in general, firms had a small number of guarantors, and did not provide guarantees to others frequently. Both in- and out-degrees exhibit a power-law distribution, with a slope ($\lambda_{in}$ and $\lambda_{out}$) ranging from 3.23 to 3.30, and 2.30 to 2.76, respectively, indicating that the guarantee network are scale-free, a property that commonly exists in real-world networks. In such a scale-free network, most firms have a small number of guarantee relationships, while a few hub firms provide/obtain guarantees to/ from many others. A closer examination of the loan guarantee data reveals that isolated mutual guarantee relationship (2-node (%)) is significantly more frequent than that would be expected for a random network generated ($p$-value = 0.002). Here, the random network ensemble is generated by the commonly adopted Directed Configuration Model (DCM), which is configured by the in-degree and out-degree sequences of the network. Please refer to (Squartini and Garlaschelli)[23] for details. The financial properties show that these isolated mutual guarantee couples are at the bottom of the whole system with low assets and credit line and high default rate, indicating that high-risk firms are more likely to obtain loans with the guarantee from firms that are also high-risk. We define two types of hubs: guarantor hubs and borrower hubs. The guarantor hubs are those giving guarantees to others most frequently (top 1% out-degree). The borrower hubs are those obtaining guarantees from others most frequently (top 1% in-degree). We find that the guarantor hubs are usually firms with large assets, liabilities, and credit line. About 15% of them are listed firms, (compared with 3.95 %, the listed ratio of the whole network). On the other hand, the borrower hubs tend to have medium amount of asset and liability, but with high default rate and risk rating. The overlap of guarantor and borrower hubs is rather small (around 15%) as compared with other real-world networks, echoing the afore-mentioned finding that guarantor hubs and borrower hubs have different characteristics.

Second, the largest weakly connected component (WCC) and the largest strongly connected component (SCC) stand for only 27.99% and 0.62% of all nodes in the network, respectively, indicating that the network is more decentralized than most other complex networks such as social networks and biological networks. Within the largest WCC and SCC, we observe the small-world effect as indicated by the small average shortest path length in the largest WCC (10.10 ~ 10.54). In addition, the average clustering coefficient (C (%)) is also relatively large (0.97% ~ 1.49%), indicating a strong transitivity effect (a friend's friend is also a friend) throughout the guarantee network ($p$-value < 0.001 as compared with the DCM null model[23]). The value is also much higher than most of the real-world complex networks[12]. Such transitivity effect shows that the guarantee is a strong trust relationship.

Third, we summarize the basic financial characteristics in Supplementary Table 1. In general, the average leverage ratio, as measured by total liabilities divided by total assets, has been maintained at around 60%, which is higher than the average leverage ratio for listed firms in the network (around 40%). The ratio of listed firms was relatively low (around 2%~5%). The ratio of listed firms decreased over time, indicating more and more guarantees were obtained by non-listed firms. This echoes the above findings of large portion of isolated mutual guarantee relationships. We will discuss this in detail in the following dynamic network analysis.

**Dynamic network analysis**. Analyzing the dynamics of the guarantee network could reveal the short- and long-term impact

of economic conditions and policies on the evolution of the credit system. We depict the topological properties and financial characteristics of the guarantee network for 63 months (01/2007–03/2012) that cover the global financial crisis and the implementation of the Chinese economic stimulus program. Figure 1 presents the evolution of the network scale. Figure 2 presents the dynamics of various topological properties of the guarantee network. Figure 3 presents the dynamics of the financial properties of firms. In these figures, the vertical lines indicate bankruptcy of NCFC (New Century Financial Co.) in April 2007, bankruptcy of LB (Lehman Brothers Inc.) in September 2008; start of the CESP (Chinese economic stimulus program) in December 2008, and end of the CESP in December 2010.

In general, the guarantee network exhibits different dynamic patterns in three phases. After the bankruptcy of New Century Financial Co., the network size increased slowly, resulting in a less densely connected network. Then, the network size dropped after the bankruptcy of Lehman Brothers Inc. On the other hand, the average asset of the firms in the guarantee network increased, indicating that the failed firms were with a relatively lower asset as compared with those did not fail. The implementation of the stimulus program immediately stopped the decrease. Since then, the network had been increasing (almost linearly), even after the end of the stimulus program. The Pearson correlations between the growth of the amount of new loans and the counts of nodes and edges are 0.56 ($p$-value < 0.001) and 0.58 ($p$-value < 0.001), respectively, indicating that the network size and amount of loan from banks are positively associated. Similar pattern was also observed for the number of WCC, and the size of the largest SCC and WCC.

We observe clear turning points during Phase 2, when the stimulus program was being implemented. More interestingly, the turning points for most topological properties were the same, but different for a few properties. In the following, we focus on the description of these turning points, and the economic implications of them.

The average degree, the exponents of the power-law in-degree distributions, reciprocity, ratio of fully connected 3-node-subgraph, and average clustering coefficient were increasing rapidly after the initiation of the stimulus program (Fig. 2). These values suddenly dropped in the last quarter of 2009, and quickly resumed the increasing trend until April 2010. Since then, all these values kept decreasing until the end of 2011 except a snap surge right after the end of the stimulus program (December 2010). The exponent of the power-law out-degree distribution followed a similar pattern, expect that it did not drop significantly in April and May 2010.

These findings indicate that there were many new mutual guarantee relationships as a result of the stimulus program. This included both mutual guarantees between two firms (reciprocity), and among three firms (ratio of fully connected 3-node-subgraph). Comparing the dynamics of the average assets and loans between the overall guarantee network and mutual guarantee network (Fig. 3), we find that these newly mutual guarantee relationships were mainly formed by firms that were with low assets and loans ($p$-value < 0.001 in chi-square tests). A closer look at these mutual guarantee firms reveals that around 70% of them were not part of the largest WCC, but in other smaller isolated components. After the initiation of the stimulus program, the inclusion of these small firms caused the surges of loans and the loans to assets ratio (as shown in Fig. 3). This trend was turned over briefly in late 2009 because of the fine adjustments of the People's Bank of China. However, the trend resumed in 2010, until People's Bank of China improved the macro-control regulations by increasing the reserve requirement ratio.

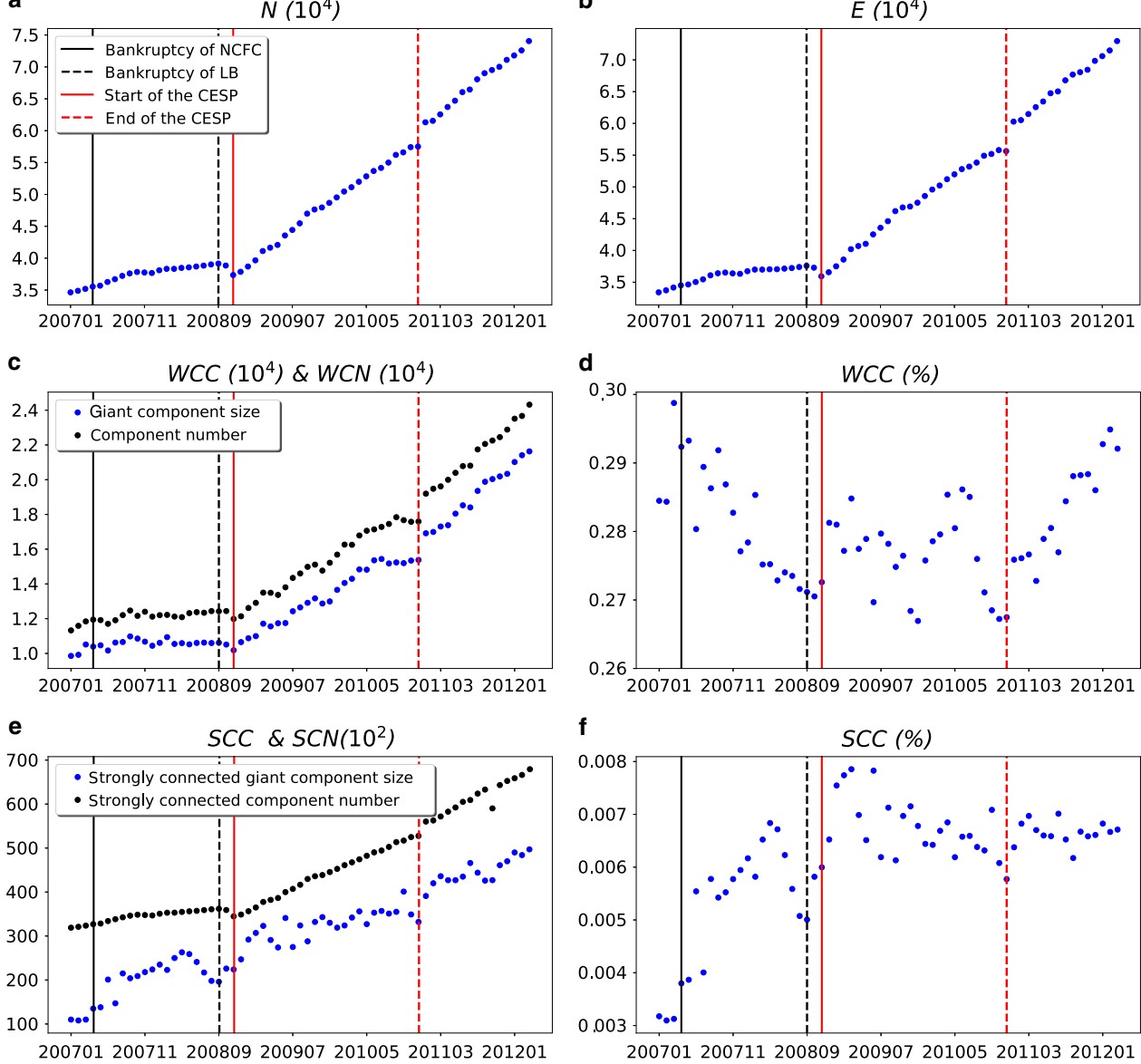

**Fig. 1 Evolution of the scale of the guarantee network. a** $N$ $(10^4)$: number of nodes. **b** $E$ $(10^4)$: number of edges. **c** $WCC$ $(10^4)$: size of the largest weakly connected component (giant component); $WCN$ $(10^4)$: weakly connected component number. **d** $WCC$ (%): ratio of weakly connected giant component. **e** $SCC$: the size of the largest strongly connected component (giant component); $SCN$ $(10^2)$: count of strongly connected component. **f** $SCC$ (%): ratio of strongly connected giant component. Source data are provided as a Source Data file.

Different from the other properties, the average shortest path length in the largest $SCC$ and $WCC$ kept increasing after the initiation of the stimulus program until September 2010. This pattern is similar to the size of the largest SCC and WCC. These findings indicated that the core of the guarantee network was not influenced significantly by the changes in regulations, until September 2010, only one month before People's Bank of China increased the interest rates. This echoes the above finding that mutual guarantee relationships were mostly in smaller isolated components.

In short, all turning points appeared in Figs. 2 and 3 are well-aligned with the changes of monetary policies and the practice of the stimulus program. At the beginning of the stimulus program, the monetary policy was ultra-loose. People's Bank of China took actions to tune the loose monetary policies in the second half of 2009. These fine tuning did not directly change the monetary policy, but aimed to address the side-effect caused by the ultra-loose monetary policy by making it more targeted, flexible and

forward-looking [goo.gl/Nq9yA5]. It was perceived as a signal to change the monetary policy. In March 2010 (17 months since the initiation of stimulus program), the government announced the plan to improve macro-control regulations in 2010. Since then, People's Bank of China increased the required reserve ratio, and raised the interest rate. Our results indicate that these regulations and policy changes, to some extent, helped to reduce the negative consequence of the stimulus program.

To check the robustness of the findings above, we constructed weighted guarantee networks with the amount of loans as the weight on edges, and did the same analyses. Please refer to the Supplementary Fig. 1 and discussions in the Supplementary Information for detailed discussions.

**Interpreting the network structure using ERGM.** The network analysis shows that the guarantee network experienced dramatic changes during financial crisis and the stimulus program. The

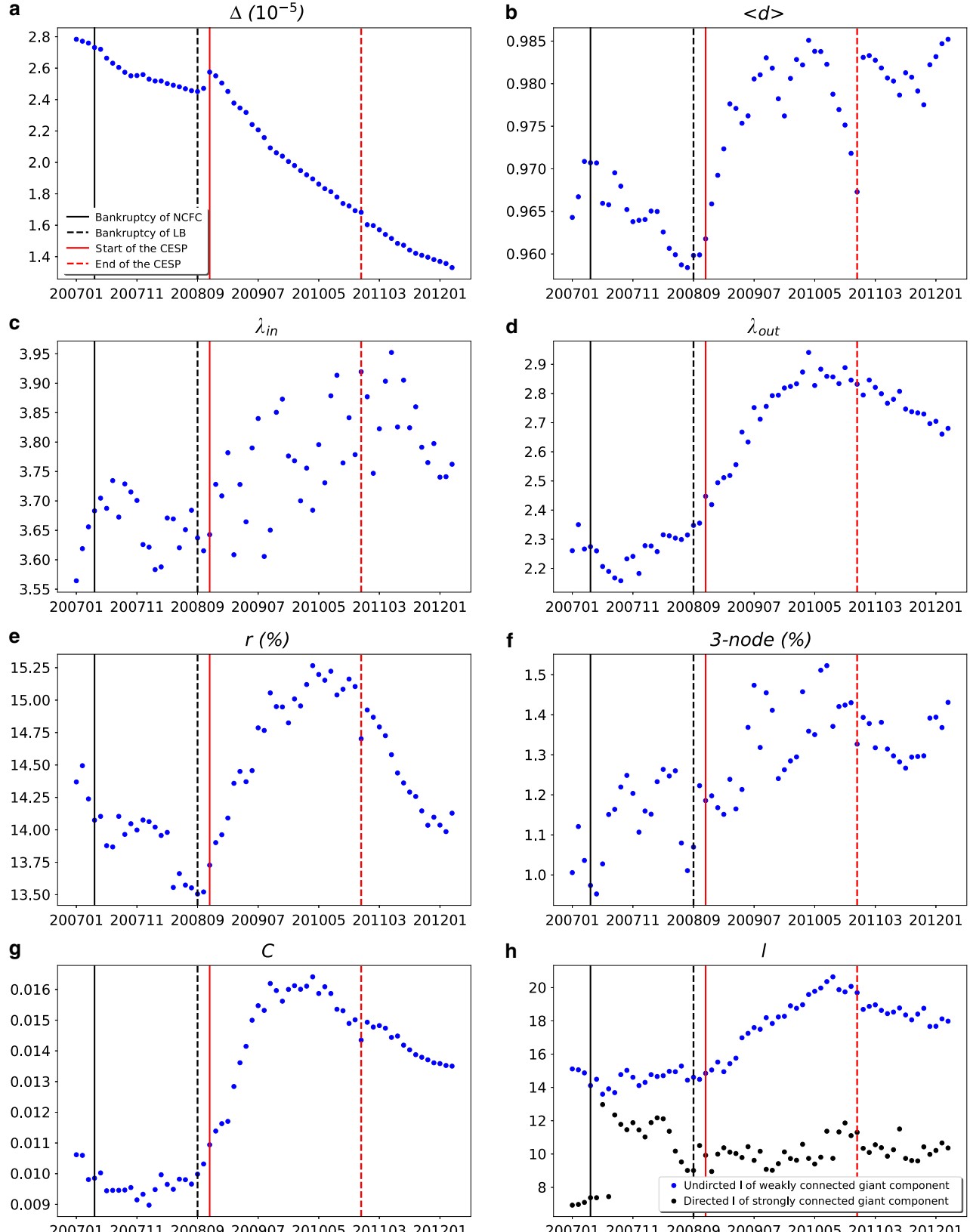

**Fig. 2 Dynamics of the topological properties of the guarantee network. a** $\Delta$ ($10^{-5}$): density. **b** $<d>$: average out/in-degree. **c** $\lambda_{in}$: power-law index of in-degree distribution. **d** $\lambda_{out}$: power-law index of out-degree distribution. **e** $r$ (%): reciprocity. **f** 3–node (%): ratio of fully connected three nodes. **g** $C$: average clustering coefficient (directed). **h** $l$: average shortest path length in the weakly connected giant component and strongly connected giant component. Source data are provided as a Source Data file.

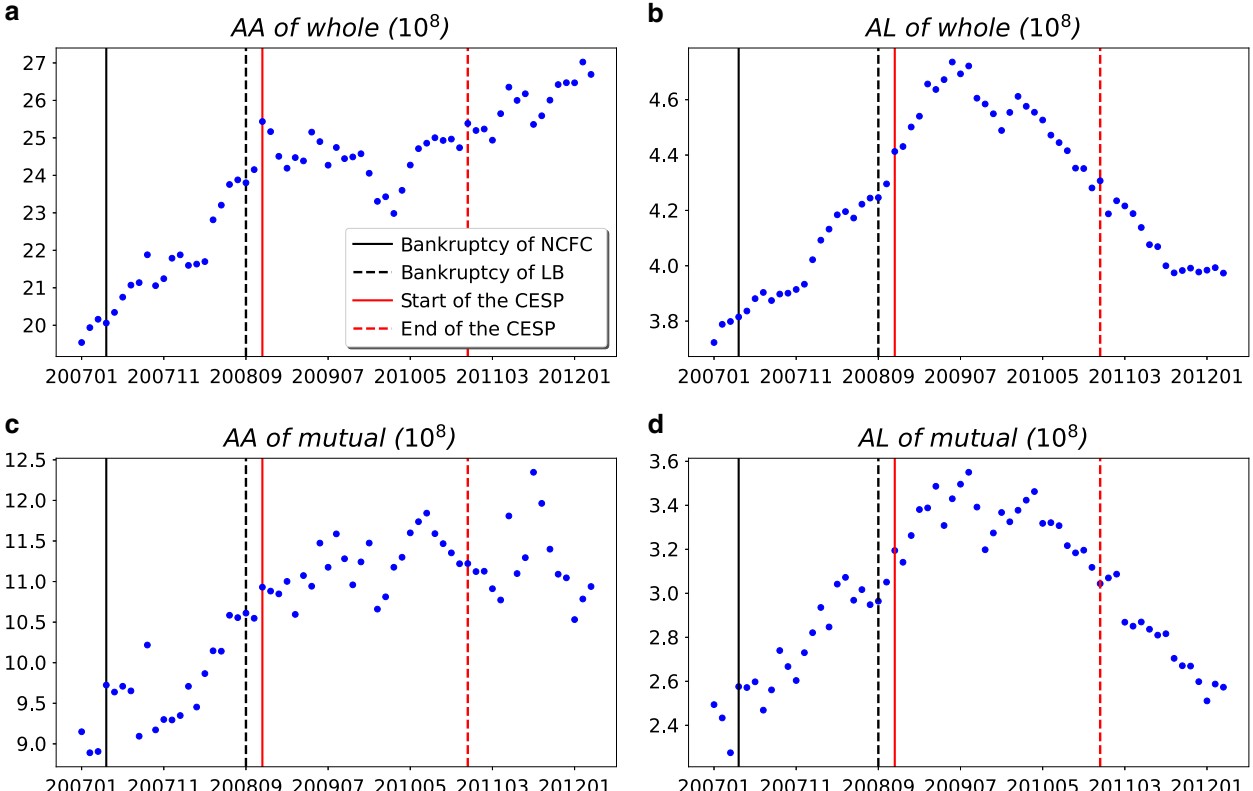

**Fig. 3 Dynamics of financial characteristics of firms in the guarantee network.** **a** AA of whole: average firms' asset of whole guarantee network. **b** AL of whole: average firms' loan of whole guarantee network. **c** AA of mutual: average firms' asset with mutual guarantee. **d** AL of mutual: average firms' loan with mutual guarantee. Source data are provided as a Source Data file.

burst of mutual guarantees could be the main factor that influences the changes of the guarantee network. In this subsection, we used exponential random graph models (ERGM)[24] to verify the significance of the formation of mutual guarantee relationships. ERGM are a family of probabilistic models that identify the distribution over a specified set of graphs that maximizes the entropy subject to the known constraints[25,26]. ERGM can effectively predict the edges as a function of these constraints, which usually represent the most important topological properties of the network structure[27]. It has been applied to analyze various networks in finance and economy, such as[28–30,23].

Among all topological properties, network density and reciprocity are the defining characteristics of the guarantee network, and other properties are with similar patterns driven by the values of these two properties. Network density is the most basic property that had dramatic changes: first increasing because of many bankruptcies, and then decreasing because of many new loans and guarantees. Reciprocity is important because the stimulus program encouraged low-quality firms to obtain loans by providing mutual guarantees to each other. In addition, we have found that the reciprocal edges are significantly more frequent than that would be expected for a DCM null model (p-value < 0.001, z-score > 3000) throughout the entire time period. Therefore, we adopt the reciprocity model[25], which considers the number of edges and the number of reciprocal edges. The probability function of $G$ is as follows:

$$P(G) = \frac{e^{\beta_1 D(G) + \beta_2 R(G)}}{Z}, \quad (1)$$

where $D(G)$ denotes the number of edges, and $R(G)$ denotes the number of reciprocal edges. $Z$ is the partition function defined by Eq. (10). Following the procedure in Methods section, the

coefficient of $D(G)$ and $R(G)$ are obtained (Fig. 4). On the other hand, the edge (representing both mutual and non-mutual guarantee relationship) is not significant. In addition, $\beta_1$ was negative and decreasing but $\beta_2$ was positive and kept increasing after the initiation of the stimulus program until March 2012, except a drop at the end of the stimulus program. These results indicate that firms were becoming increasingly likely to form mutual guarantee relationship. The ERGM analysis further demonstrates the significant role of mutual guarantee in the formation of the guarantee network.

**Simulation analysis of the resilience of guarantee network.** Network analysis reveals that the stimulus program dramatically changed the topological properties of the guarantee network, and resulted in a surge of mutual guarantee relationships. The following adjustments of monetary policies constrained the guarantee behaviors, thus reverted the increasing trend of mutual guarantee relationships. Such mutual guarantee relationship, though helped low-quality SOEs obtain more loans to survive, could result in large-scale cascading failures/defaults because of the risk and failures propagates via the guarantee relationships among firms. In this section, we examine the potential risk of cascading failures in the guarantee network using a simulation model.

We represent the assets and liabilities of a firm $i$ at time $t$ as $A_i(t)$ and $L_i(t)$, respectively. $G_{ij}(t)$ denotes the amount of loan guarantee firm $i$ provides for firm $j$ at time $t$. Following the literature, we adopt the Fermi distribution model to calculate the probability of default failure[31,32]. Fermi distribution is a logistic function that has been successfully applied to modeling failures of firms and banks in economics and finance[31,32]. In our research, $P_i(t)$ is a Fermi logistic function that determines the probability

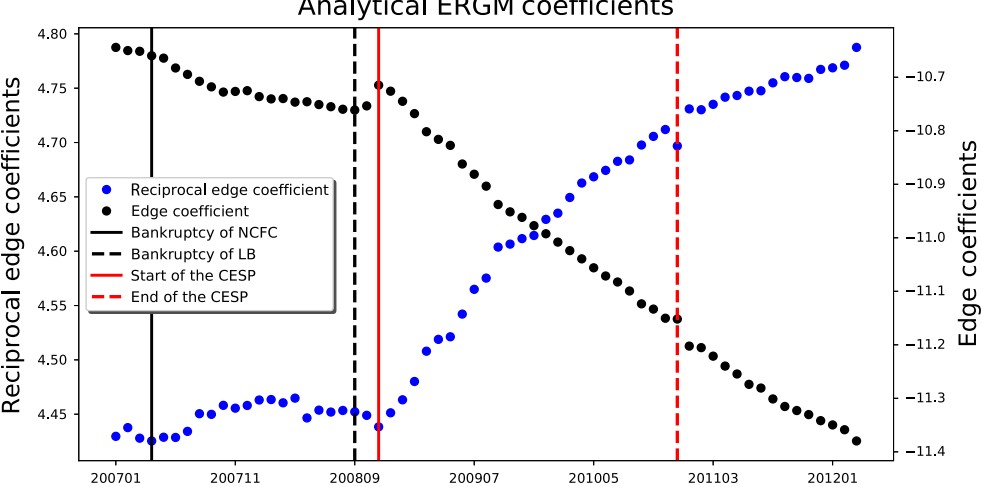

**Fig. 4 Dynamic changes of coefficients in ERGM.** Source data are provided as a Source Data file.

that a firm defaults at time $t$:

$$P_i(t) = \frac{1}{1 + e^{-k\left(\frac{L_i(t) + \sum_j w_j(t)G_{ij}(t)}{A_i(t)} - \delta\right)}}, \qquad (2)$$

where $k$ represents the influence of external environment, $\delta$ represents the mean value of the leverage ratio (total liabilities/total assets), and $w_j(t)$ represents whether firm $j$ has failed. The leverage ratio is determined by the current assets and liabilities of a firm, as well as the assumed debt caused by the failure of the firms to which $i$ provides guarantee. Intuitively, the higher the leverage ratio a firm has, the more likely it would fail.

The simulation is based on the aforementioned dynamic guarantee network and the financial characteristics obtained from real-world data. We simulate the failures of firms caused by random initial failures (random attack) in one month. In the simulation, $\delta$ is the average leverage ratio of firms from real-world monthly guarantee network data. $k = 1.18$, which is obtained by learning from the real data. Specifically, we adopt Eq. (2) as a logistic regression to fit the real values of $P_i(t)$. Specifically, if firm $i$ at time $t$ defaulted, $P_i(t) = 1$, otherwise $P_i(t) = 0$. The values of other factors, including $A_i(t)$, $L_i(t)$, $G_{ij}(t)$, and $\delta$ of firm $i$, at time $t$ were provided by our dataset. $\eta \in (0,1)$ represents the initial default ratio. In particular, for each month, the simulation runs as follows. Initial failures: At the beginning of each simulation ($t = 1$), a small proportion ($\eta = 0.05$) of firms were randomly selected as the initial failures (default). Their liabilities will be assumed by the adjacent firms in the network. Failure propagations: At each time step $t$, we determine whether a firm $i$ fails or not at time $t$ based on Eq. (2). Finish: The simulation keeps running until there is no firm fails at $t_{\text{finish}}$, $t_{\text{finish}} \geq 1$.

We run the simulation for 10,000 times for each month, and take the average ratio of failed firms for each month (Fig. 5). We find that the dynamic pattern of average ratio of failed firms is similar to the common pattern of topological properties (average degree, the exponent of the power-law in-degree distributions, average reciprocity, average ratio of fully connected 3-node-subgraph, and average clustering coefficient). The correlation between the average ratio of failed firms and the average reciprocity is very high (correlation coefficient = 0.83, $p$-value < 0.001). The turning points are the same (September 2009 and April 2010). This result indicates that the more prevalent mutual guarantee relationships are, the more fragile the whole guarantee network is. Please also check the robustness check of the simulation analysis in the Supplementary Note 4.

To summarize, the stimulus program caused an elevated risk of cascading failure in the credit system. The fragility of the guarantee network was strongly influenced by the stimulus program and changes of monetary policies. The follow-up adjustment reduced such risk, but the risk was still higher than it was before the financial crisis. Clearly, these findings do not coincide with our traditional understanding of systemic crisis development. According to general assumption, the stability of financial network should become worse during financial crisis, and the government bailout of stimulus program could reduce the systemic risk and restore the network stability. Nonetheless, the above findings lead us to opposite conclusions: the 2007–2008 financial crisis resulted in more stable guarantee network, and the stimulus program decreased the stability of guarantee system. Such counterintuitive findings can be explained from the following perspective of survivorship bias.

In the real world, the external negative shock from financial crisis would knock out the low-quality enterprises and fragile linkages, resulting in a smaller and less connected network. Those who were left behind are high-quality enterprises with robust guarantee relationships, leading to a more stable guarantee network. When it came to the government's rescue, with the injection of RMB ¥4 trillion into the credit market, the stimulus program saved substantial low-quality enterprises on the verge of bankruptcy. As a result, the enterprises that survived by chance increased the network connectivity with more fragile links, thus lowering the stability of the credit guarantee network. In short, the behavior of survivors within financial crisis and stimulus program played a crucial role in the network stability.

## Discussion

This research presents the first attempt to quantitatively characterize the evolution of the entire Chinese guarantee system as a complex network. The empirical and simulation studies provide evidence that the financial crisis caused a large number of weak firms unqualified to access bank loan, making the guarantee network smaller, and the loose monetary policy along with the stimulus program encouraged the mutual guarantee behavior between firms, resulted in highly reciprocal and fragile network structure. The latter adjustment of the monetary policy reduced the ratio of mutual guarantee relationships, and enhanced the resilience of the whole guarantee network.

Our study contributes to the literature through proposing a complex network approach to analyzing the guarantee relationships between firms in a whole country. The simulation model

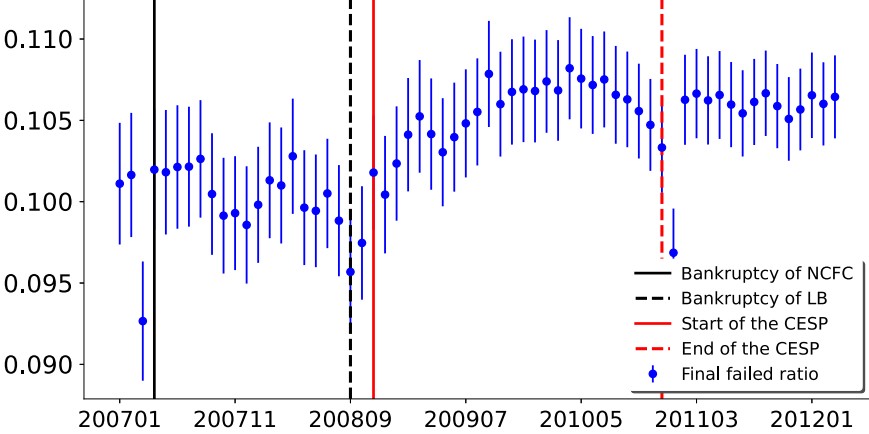

**Fig. 5 Final failed ratio of random attack with the initial default ratio $\eta$ equal to 0.05.** The vertical axis represents the final failed ratio. The horizontal axis represents time from January 2007 to March 2012. The solid and dashed lines represent major events. In each month, 5% nodes are randomly selected as the initial failures (default). The error bar represents one standard deviation. Source data are provided as a Source Data file.

based on real-world guarantee network presents a novel network-based method to evaluate the fragility of the guarantee systems. The empirical and simulation results provide data-driven insights of how the systemic risk, government bailout and following monetary policy adjustments influence the guarantee behaviors of firms, and the fragility of the guarantee system. This study also sheds light on the data-driven research on using the network science methodologies to analyze real-world financial systems. The insights into the evolution of the guarantee network have great potential to be applied to other finance and economic research, such as forecasting stock market returns[33], credit allocation[7] and international trade[23].

In practice, this study indicates that although the mutual guarantee relationship could help low-quality firms obtain more loans, and reduce the risk of small-scale arbitrary risk events, it could be the cause of large-scale cascading failures/defaults for the whole system. This research suggests that decision makers (e.g., government, central bank, and other authorities) control the prevalence of mutual guarantee relationships through adjusting monetary policies.

Importantly, much work about resilience of financial systems has focused on the theoretical analysis due to the unavailability of confidential financial data. Therefore, the real conditions and evolution of financial systems have not yet been fully understood. Here, with the nation-wide and real-world Chinese loan guarantee data, we show that the empirical results are opposite to the intuitions. Specifically, to take the effect of survivorship bias into consideration, the guarantee network is much more stable during the period of financial crisis with the surviving and robust firms, however, the stability keeps on decreasing with the implementation of stimulus program, for absorbing much more firms which should have been bankrupted without this government bailout. Our empirical findings would facilitate the study of stability of real-world financial system and also enable better strategies to be developed for policy makers to perform financial bailout.

The results of this study also pose an interesting dilemma. Financial crisis has made the financial network healthier by obsoleting the weak firms, but with instantaneous huge economic consequence. The government bailout avoided the instantaneous catastrophic loss at the expense of making the financial network fragile. In other words, government bailout could mitigate the current crisis by sacrificing the financial stability in the future. The 2019-2020 novel coronavirus pandemic in fact tells a similar story. Risk management needs a consecutive endeavour that

considers such trade-offs in a proactive manner. The methods and results of this study provide such benchmark data, simulation platform, and baseline methods to facilitate decision makers in developing effective responses to crisis.

Although this study relies on the data of the Chinese guarantee network, the results and implications are still very beneficial from three perspectives. First, China is the second largest economy in the world. The credit system is unique in many aspects (e.g., the significant role of indirect financing and bank credit in the credit system) as compared with those in many developed countries, making it important to examine how the uniqueness of the Chinese credit system is associated with the unique challenge in running the financial system, especially in crisis. Second, the credit guarantee relationship among Chinese firms are similar to those of other countries, as they are all following the standard procedure of commercial banks. Thus, the guarantee network is general, as well as the network science methodologies. If we have equivalent data for another credit system, the methods are directly applicable by calibrating the parameters of guarantee network and simulation model based on the new data. Many topological properties could be similar, such as the scale-free and small-world effects. In practice, it is very difficult to obtain similar nationwide guarantee system data. In such case, we still can use the topological and financial properties of the Chinese guarantee network as a baseline, and then calibrate the parameters to reflect our understanding of the guarantee system under consideration. For example, we may reduce the reciprocity to reflect the less prevalence of mutual guarantee behaviors in a non-Chinese credit system. Third, we have been witnessing crises frequently, particularly the 2019–2020 novel coronavirus pandemic, which may trigger another financial crisis in the near future. The results of this study provide needed data-driven insights into how the credit system may react and recover. Such insights are critical to the decision making not only in China, but elsewhere as well.

## Methods
**Data**. In this research, we acquire a comprehensive nationwide dataset from one of the major regulatory bodies in China. The data spans from January 2007 to March 2012, and contains the monthly information for all loans extended to the client firms, which have a credit line above 50 million RMBs. These loan guarantee data are from all the 19 major banks of China, including the largest five state-owned commercial banks, twelve joint equity commercial banks, and two major policy banks (Import-Export Banks of China and China Development Bank), which account for nearly 80% total loans in China. Totally, there are about 0.3 million guarantee relationships during 01/2007–03/2012, which covers around 0.1 million

**Fig. 6 Illustration of the dynamic loan guarantee relationships in the guarantee network.** In time $t$, there are three guarantee relationships. Firm A was the guarantor for borrowers B and C. Firm E was the guarantor for borrower C. In time $t+1$, the guarantee relationship between A and C was gone after C repaid the loans. Firm D became the guarantor for A, who was both a guarantor and a borrower. In time $t+2$, the guarantee relationship between C and E was gone. Firm E became the guarantor for firm D. Firm A became the guarantor for firm D, which is also the guarantor for firm A. Firms A and D formed the mutual guarantee relationship (reciprocal edge).

borrowing firms located in 30 province-level regions of China. More specifically, the data contains details regarding loan-level information for each loan guarantee (borrower, provider, as well as amount of each loan guarantee, and the time of the guarantee relationship) and firm-level fundamentals (e.g., size, leverage, and location). To the best of our knowledge, our data is by far the largest and the most comprehensive representation of the Chinese credit system.

Particularly, our dataset covers two important periods in the credit system: The 2007–2008 global financial crisis, and the implementation of the Chinese economic stimulus program from November 2008 to December 2010. Through harnessing our data, these two rare natural experiments offer a good opportunity for us to investigate the change of guarantee network under extreme exogenous shocks.

**Construction and analysis of the guarantee network.** To analyze the structure and dynamics of Chinese guarantee system, we constructed dynamic guarantee networks using the guarantee relationships between firms. Figure 6 illustrates the construction of the networks. Each node represents a firm, and each edge connecting two nodes represents the existence of guarantee relationship between the two corresponding firms in a specific month. An edge goes from the guarantor to the borrower. This dynamic guarantee network consists of 63 monthly loan guarantee data, enabling us to analyze the evolution and dynamics of the system over time.

In this research, we adopted a set of commonly used topological metrics to characterize the structure and evolution of the guarantee network:

**Network connectivity** refers to how well nodes are connected with each other in the network. It is measured by three metrics, number of weakly or strongly connected components (WCN or SCN), ratio of the giant component (the largest weakly connected component, or largest WCC (%)), in the whole network, the ratio of the largest strongly connected component, or largest SCC (%), in the whole network, and the network density ($\Delta$)[34,35]. In a weakly connected component (WCC), any node can reach another node through an undirected path. The giant component is the weakly connected component with the largest number of nodes. Similarly, in a strong connected component (SCC), any node can reach another node through a directed path.

Network density is the ratio between the number of directed edge and the total number of possible directed edges, density $= \frac{E}{N(N-1)}$, where $E$ is the number of edges and $N$ is the number of nodes in the network.

Degree, $<d>$, refers to the number of edges adjacent to a node. In the guarantee network, edges are with directions, from guarantor to borrower. The degree of a node indicates the extent to which it is connected within the system. In-degree measures the number of guarantors for the firm. Out-degree measures the number of borrowers that received guarantee from the firm. In general, a higher value of average out/in-degree indicates more frequent guarantee relations among firms in the network.

Scale-free property has been observed in many real-world networks. In a scale-free network, degree distribution follows a power-law. Most nodes are only connected to a small number of edges, while there exist a few hub nodes that are densely connected. To test the scale-free property of the guarantee network, we investigate both the in- and out-degree distributions, denoted by $p_{in}(k)$ and $p_{out}(k)$. In a scale-free network, both $p_{in}(k)$ and $p_{out}(k)$ follow a power-law distribution. The frequency of nodes with degree of $k$ is proportional to $k$ to the power of $\lambda$, $p_{in}(k) \sim k^{-\lambda}$ and $p_{out}(k) \sim k^{-\lambda}$[36].

Clustering coefficient, $C(\%)$, measures the extent to which a node's neighbors are also adjacent to each other. In real-world networks, nodes are likely to form such triads, resulting in high average clustering coefficient of the networks. In social networks, it refers to the tendency that friend of a friend is also a friend[37,38]. In the guarantee network, high clustering coefficient indicates that firms tend to form tightly guarantee clusters with high frequency of guarantees among them.

The reciprocity, $r(\%)$ (dyad; mutual guarantee relationship between two firms) of a directed network is the ratio of the number of reciprocated edges to the total number of edges. Here, node $A$ and node $B$ are reciprocated if an edge is connected

from node $A$ to node $B$, and there is also an edge from node $B$ to node $A$. In the guarantee network, it measures the ratio of mutual guarantee relationships to the total number of guarantee relationships in the network.

Ratio of fully connected 3-node-subgraph, 3–$node$ (%) (triad; mutual guarantee relationship among three firms) of a directed network is the ratio of the number of edges forming a fully connected 3-node-subgraph to the total number of edges in the graph. In the guarantee network, it measures the extent to which the three firms provide mutually guarantees to each other.

Ratio of isolated 2-node reciprocal component, 2–$node$ (%), measures the prevalence of isolated 2-node mutual component in the network. It is calculated by dividing the number of isolated 2-node reciprocal component with the total number of weakly components in the network. This measure quantifies the extent to which two firms formed mutual guarantee relationships without any interaction with other firms.

We measure the efficiency of the guarantee network in transferring the risk by calculating the average shortest path length, $l$, of the network. The average shortest path length is defined as the average number of edges along the shortest path connecting all possible pairs of nodes. Real-world networks usually exhibit a relatively small average shortest path length, indicating the small-world property[39–41]. Because the guarantee network has multiple connected components, we calculate the average shortest path length of the largest weakly connected component and the largest strongly connected component.

**Financial property.** We calculate a set of financial properties, including average assets and average loans of firms in the guarantee system, denoted by $AA$ and $AL$, respectively.

More specifically, the average assets $AA$ refers to average amount of assets of all firms in the guarantee system. A large value of $AA$ indicates that the guarantee system is consisted of firms with high values.

The average loans $AL$ refers to the average amount of loans of all firms in the guarantee system. A large value of $AL$ indicates that the guarantee system is consisted of firms that owe a lot of debts to banks.

We further calculate the average ratio of listed firms, $ARL$ (%), which is denoted $ARL = \frac{\text{number of listed firm}}{\text{number of all firms}}$. A large value of $ARL$ indicates that the guarantee system has more firms whose stock trades on a stock exchange.

**Exponential random graph models.** Given the structure of real network, first we need to choose a set of topological properties as constraints, denoted as $C_i^* (i = 1, 2, 3 \ldots k)$, where $k$ is the total number of constraints[23]. We consider a set of networks $\mathcal{G}$, an ensemble of all networks of $n$ nodes without self-loops, whose expected value of constraints $<C_i>$ over $\mathcal{G}$ is equal to that of the real guarantee network $(C_i^*)$. The probability of a network $G \in \mathcal{G}$ is denoted as $P(G)$. It has been proved that we can obtain the value of $P(G)$ under constraints $C_i^*$ by maximizing the Gibbs entropy $S$, which is defined as follows,

$$S = -\sum_{G \in \mathcal{G}} P(G) \ln P(G),$$ (3)

with the constraints

$$<C_i> = \sum_{G \in \mathcal{G}} P(G) C_i(G) = C_i^*,$$ (4)

and the normalization condition

$$\sum_{G \in \mathcal{G}} P(G) = 1,$$ (5)

where $C_i(G)$ is the value of $C_i$ in network $G$.

Using the Lagrange multipliers, we can find that the maximum entropy is achieved for the distribution satisfying

$$\frac{\partial}{\partial P(G)}\left[S - \alpha\left(1 - \sum_{G\in\mathcal{G}} P(G)\right) - \sum_i \beta_i\left(<C_i> - \sum_{G\in\mathcal{G}} P(G)C_i(G)\right)\right] = 0. \quad (6)$$

By solving Eq. (6), we obtain

$$-\ln P(G) - 1 + \alpha + \sum_i \beta_i\, C_i(G) = 0. \quad (7)$$

Or, equivalently, the probability of graph $G$ is obtained as follows

$$P(G) = \frac{e^{H(G)}}{Z}, \quad (8)$$

with the graph Hamiltonian function

$$H(G) = \sum_i \beta_i\, C_i(G), \quad (9)$$

and the partition function

$$Z = e^{1-\alpha} = \sum_{G\in\mathcal{G}} e^{H(G)}. \quad (10)$$

We define the topological structure of a network $G$ with an adjacency matrix $\mathbf{A}$, whose entries $a_{ij}=1$ if node $i$ and node $j$ are connected, and $a_{ij}=0$ otherwise. In our study, we denote the real guarantee network by the particular matrix $\mathbf{A}^*$.

In this study, we chose two constraints: edge number and reciprocal edge number of guarantee network. Please refer to Results section for discussions of why these two properties are the defining features of the Chinese guarantee network. We assume that the expected number of edges $<D(G)>$ and the expected number of mutual edges $<R(G)>$ of the real network are known. Then, the Hamiltonian of $G$ is

$$H(G) = \beta_1 D(G) + \beta_2 R(G), \quad (11)$$

where $D(G) = \sum_{i<j}(a_{ij} + a_{ji})$ denotes the number of edges, and $R(G) = 2\sum_{i<j} a_{ij}a_{ji}$ denotes the number of reciprocal edges. Given fixed number of nodes, $R(G)$ and $R(G)$ essentially evaluate the density and reciprocity of the network given the fixed number of nodes in the network, respectively.

Taking (11) into (10), the partition function for the complete system is formulated as

$$\begin{aligned} Z &= \sum_{\{a_{ij}\}}\sum_{\{a_{ji}\}} \exp\left(\sum_{i<j}[\beta_1\,(a_{ij} + a_{ji}) + 2\beta_2\,a_{ij}a_{ji}]\right) \\ &= \prod_{i<j}\sum_{a_{ij}=0,1}\sum_{a_{ji}=0,1} e^{\beta_1(a_{ij}+a_{ji})+2\beta_2\,a_{ij}a_{ji}} \\ &= \prod_{i<j}\left[1 + 2e^{\beta_1} + e^{2(\beta_1+\beta_2)}\right] = \left[1 + 2e^{\beta_1} + e^{2(\beta_1+\beta_2)}\right]^{\binom{n}{2}}. \end{aligned} \quad (12)$$

Then, the free energy of network is

$$F = \ln Z = \binom{n}{2}\ln\left(1 + 2e^{\beta_1} + e^{2(\beta_1+\beta_2)}\right). \quad (13)$$

Finally, the expected values of edge and reciprocal edge are

$$<D(G)> = \frac{\partial F}{\partial\beta_1} = n(n-1)\frac{e^{\beta_1} + e^{2(\beta_1+\beta_2)}}{1 + 2e^{\beta_1} + e^{2(\beta_1+\beta_2)}}, \quad (14)$$

$$<R(G)> = \frac{\partial F}{\partial\beta_2} = n(n-1)\frac{e^{2(\beta_1+\beta_2)}}{1 + 2e^{\beta_1} + e^{2(\beta_1+\beta_2)}}. \quad (15)$$

We set the expected values of edge and mutual edge to be equal to the actual counts of edges and reciprocal edges, respectively. From Eqs. (14) and (15), we can obtain the values of $\beta_1$ and $\beta_2$, which represents the log-odds of the probability of edges and the number of reciprocal edges, respectively. Then, the probability of edges and the probability of reciprocal edges are the inverse-logits of $\beta_1$ and $\beta_2$, respectively. Thus, a large value of the coefficient indicates the high probability of forming the corresponding topology (edge and reciprocal edge). Note that the coefficient can be negative, indicating the low prevalence of such topology in the real network. For more details of interpretations of ERGM, please refer to[23,25].

**Reporting summary**. Further information on research design is available in the Nature Research Reporting Summary linked to this article.

## Data availability
The dataset used in the paper is confidential and only available to reader on request. The source data underlying all figures, tables are provided as a Source Data file.

## Code availability
The source codes have been made available on Github: [https://github.com/ElfLu/Chinese-credit-network-.git]

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

## Acknowledgements

This work was supported by National Natural Science Foundation of China Grants 71532013, 71850008, 71972164, and 71672163.

## Author contributions

The three authors are joint first authors. Y.W., Q.Z., and X.Y. designed research; Y.W., and Q.Z. analyzed data and performed research; and Y.W., Q.Z., and X.Y. wrote the paper, and worked on the revision.

## Competing interests

The authors declare no competing interests.
