## [Peer Review File · Nature Communications]

Reviewers' comments:

Reviewer #1 (Remarks to the Author):

The authors analyse the Chinese guarantee network in the aftermath of the global financial crisis and find that the stimulus program promoted by the Chinese government after 2008 had an impact on the evolution of the considered system during the biennium 2009-2011.

Although the topic is timely and authors' premises are correct ("the systemic risk is the key to the study of financial networks but we still do not have sufficient knowledge about the systemic risk of real-world financial networks due to the lack of comprehensive financial data about the whole system"), the analysis they carry out is far from being exhaustive, for the following reasons:

- on the one hand, the potential outcome of the authors' results is limited by the nature of the study they carried out (as the authors themselves recognize, the Chinese market is quite unique, hence my question: how generalizable are the results concerning such a system?);

- on the other hand, the core part of the analysis is merely empirical. In order to enrich their contribution, I suggest the authors to explore (and properly acknowledge) the literature about the application of Exponential Random Graph models to the analysis of economic and financial networks. More precisely, the ERG they employ (eq. 2) is the Reciprocity Model, originally proposed in J. Park and M. E. J. Newman, *Phys. Rev. E* 70, 066117 (2004). It is analytically solvable: hence, there is no need to implement numerical techniques as the MCMC. More refined (but still analytically solvable) null models have been also proposed (e.g. the Reciprocal Configuration Model): see D. Garlaschelli and M. I. Loffredo, *Physical Review E* 73, 015101(R) (2006).

In what follows, examples of the issues that the authors should clarify are provided:

- the techniques mentioned above are employed in a rather limited way. As an example, ERG models can be used to test the significance of a given set of observations (see also T. Squartini and D. Garlaschelli, *J. Complex Netw.* 3(1), 1-21 (2015)): in the present contribution, however, no significance analysis of the topological quantities of interest (i.e. the ones shown in figs. 2 and 3) has been carried out. It is only mentioned once that "A closer examination of the loan guarantee data revealed that isolated mutual guarantee relationship (2-node(%)) displays in significantly higher frequencies than would be expected for a random network ($p_{\text{value}}=0.0017$)." but without further specifying which random model has been tested and the details of the procedure that has been implemented to calculate the aforementioned statistical significance;

- the authors claim that "The empirical results of the ERGM model showed that both the density and reciprocity were significant ($p\text{-value}<0.001$) throughout the entire time period, as shown in Figure 5." What the authors mean is unclear since, by definition, the empirical density and reciprocity values are reproduced by the model: hence, their value cannot be significant in the usual, statistical sense;

- the authors say that "[...] the average clustering coefficient (C (%)) is also relatively large (0.97%~1.49%), indicating a strong "a friend's friend is also a friend" effect throughout the guarantee network, implying guarantee is a strong trust relationship.": "strong" with respect to what? Has a random model been chosen as a benchmark or other real-world networks are being considered for comparison?

- the authors just consider the weakly connected component of the network under analysis; such a network, however, is directed: what about the behavior of the strongly connected component?

- the parameter k in equation 3 is set empirically to 1.5: why? Choices like these should be properly motivated.

To sum up, while the authors are right in stating that network theory can greatly help (and greatly helped) in understanding the stability of financial systems, the proposed analysis is quite limited: as a consequence, this manuscript cannot be judged as suitable for publication in a journal like Nature Communications.

Reviewer #2 (Remarks to the Author):

In this manuscript, based complex network, the authors analyze the structure and evolution of the Chinese guarantee network with five years' worth of real-world data from January 2007 to March 2012. The authors characterize the global topological properties of the guarantee network, and find that the global financial crisis during 2007-2008 and economic policies in the aftermath (i.e., Chinese economic stimulus program and latter adjustments) had significant influence on the evolution of guarantee network structure.

The main innovations are: the authors study on the risk of a comprehensive nationwide guarantee network and the influence of economic situation and national economic policies on the topological structure of guarantee network. Taking the 2007-2008 financial crisis and the subsequent Chinese economic stimulus program as two perfect natural experiments to investigate the influence of economic situation and national economic policies on the structure of guarantee network, as well as the associated contagion risk in the system.

This manuscript is well written and the idea is interesting. But I still have some additional comments.

(1).The authors explained the research motivation of this manuscript mainly from a data-driven perspective. I suggest that the authors give more analysis on the research problems.

(2). To analyze the structure and dynamics of Chinese guarantee system, the authors constructed dynamic guarantee networks using the guarantee relationships between firms. Each node represents a firm, and each edge connecting two nodes represents the existence of guarantee relationship between the two corresponding firms in a specific month. Does the author distinguish the size of the guarantee amount? As we know, large guarantee amounts should have a greater impact.

(3). The authors investigated the influence of 2007-2008 global financial crisis and China's stimulus program in the topological structure of Chinese guarantee network. What is the network topology of foreign companies or foreign banks in China from January 2007 to March 2012?

(4). Markov chain Monte Carlo (MCMC) method was adopted to estimate the parameters of exponential random graph model. Please give some details about the algorithm.

(5).I suggest the author give corresponding implications from the trade-off between risk and return. If the next major economic and financial crisis breaks out, should the Chinese government adopt a grand economic stimulus plan?

Finally, evolution of the Guarantee Network can be applied in stock market returns problems. I suggest that the authors pay attention to the following paper:

1. Dai Z., Zhu H. (2019) Forecasting stock market returns by combining Sum-of-the-parts and ensemble empirical mode decomposition, Applied Economics, doi:10.1080/00036846.2019.1688244. Additionally, there are some typos in the manuscript, e.g.,

(1) At page 4, "In section 2," should be "In Section 2,". Please check all of this manuscript.

(2) At the bottom of page 6, "both $p_{in}(k)$ and $p_{out}(k)$ follows" should be " both $p_{in}(k)$ and $p_{out}(k)$ follow ".

(3) Need page numbers.

Reviewer 1's comments and our responses:

Comment 1-1: *The authors analyse the Chinese guarantee network in the aftermath of the global financial crisis and find that the stimulus program promoted by the Chinese government after 2008 had an impact on the evolution of the considered system during the biennium 2009-2011.*

Although the topic is timely and authors' premises are correct ("the systemic risk is the key to the study of financial networks but we still do not have sufficient knowledge about the systemic risk of real-world financial networks due to the lack of comprehensive financial data about the whole system"), the analysis they carry out is far from being exhaustive, for the following reasons:

on the one hand, the potential outcome of the authors' results is limited by the nature of the study they carried out (as the authors themselves recognize, the Chinese market is quite unique, hence my question: how generalizable are the results concerning such a system?);

Response 1-1: Thank you very much for your encouragement and very constructive comments!

We agree with the reviewer that the Chinese market is unique, and it is important to discuss the generalizability of our results. We'd like to respond to the comment from three perspectives.

- First, China is the second largest economy in the world. The credit system is unique in many aspects (e.g. the significant role of indirect financing and bank credit in the credit system) as compared with those in many developed countries, making it important to examine how the uniqueness of the Chinese credit system is associated with the unique challenge in running the financial system, especially in crisis.
- Second, the credit guarantee relationship among Chinese firms are similar to those of other countries, as they are all following the standard procedure of commercial banks. Thus, the guarantee network is general, as well as the network science methodologies. If we have equivalent data for another credit system, the methods are directly applicable by calibrating the parameters of guarantee network and simulation model based on the new data. Many topological properties could be similar, such as the scale-free and small-world effects. In practice, it is very difficult to obtain similar nationwide guarantee system data. In such case, we still can use the topological and financial properties of the Chinese guarantee network as a baseline, and then calibrate the parameters to reflect our understanding of

the guarantee system under consideration. For example, we may reduce the reciprocity to reflect the less prevalence of mutual guarantee behaviors in a non-Chinese credit system.

- Third, we have been witnessing crises frequently, particularly the 2019-2020 novel coronavirus pandemic, which may trigger another financial crisis in the near future. The results of this study provide needed data-driven insights into how the credit system may react and recover. Such insights are critical to the decision making not only in China, but elsewhere as well.

We have added the discussions of the generalizability of the results in Discussions.

Page 14-15. Although this study relies on the data of the Chinese guarantee network, the results and implications are still very beneficial from three perspectives. First, China is the second largest economy in the world. The credit system is unique in many aspects (e.g. the significant role of indirect financing and bank credit in the credit system) as compared with those in many developed countries, making it important to examine how the uniqueness of the Chinese credit system is associated with the unique challenge in running the financial system, especially in crisis. Second, the credit guarantee relationship among Chinese firms are similar to those of other countries, as they are all following the standard procedure of commercial banks. Thus, the guarantee network is general, as well as the network science methodologies. If we have equivalent data for another credit system, the methods are directly applicable by calibrating the parameters of guarantee network and simulation model based on the new data. Many topological properties could be similar, such as the scale-free and small-world effects. In practice, it is very difficult to obtain similar nationwide guarantee system data. In such case, we still can use the topological and financial properties of the Chinese guarantee network as a baseline, and then calibrate the parameters to reflect our understanding of the guarantee system under consideration. For example, we may reduce the reciprocity to reflect the less prevalence of mutual guarantee behaviors in a non-Chinese credit system. Third, we have been witnessing crises frequently, particularly the 2019-2020 novel coronavirus pandemic, which may trigger another financial crisis in the near future. The results of this study provide needed data-driven insights into how the credit system may react and recover. Such insights are critical to the decision making not only in China, but elsewhere as well.

Comment 1-2: *on the other hand, the core part of the analysis is merely empirical. In order to enrich their contribution, I suggest the authors to explore (and properly acknowledge) the literature about the application of Exponential Random Graph models to the analysis of economic and financial networks.*

More precisely, the ERG they employ (eq. 2) is the Reciprocity Model, originally proposed in J. Park and M. E. J. Newman, Phys. Rev. E 70, 066117 (2004). It is analytically solvable: hence, there is no need to implement numerical techniques as the MCMC. More refined (but still analytically solvable) null models have been also proposed (e.g. the Reciprocal Configuration Model): see D. Garlaschelli and M. I. Loffredo, Physical Review E 73, 015101(R) (2006).

Response 1-2: Thank you very much for the valuable comments! Following the reviewer's comment, we summarize and cite the application of ERGM to the analysis of economic and financial networks. We also updated the introduction about ERGM to include more recent references in the field.

Following the reviewer's suggestion, we adopted an analytical approach introduced in (Park & Newman, 2004) to replace MCMC to derive the values of coefficients in ERGM. We also added detailed steps of analytically deriving the values of coefficients and associated probabilities. We have used the Directed Configuration Model (DCM) as the null model, which is defined by the in-degree and out-degree sequences (Squartini & Garlaschelli, 2015). We did not directly use the Reciprocal Configuration Model, which fixes the reciprocity, because we are examining if mutual guarantee relationship (reciprocal edge) is the key characteristic of the network. We copied the revised parts and figure below:

The added references are copied below:

23. Squartini, T. & Garlaschelli, D. Stationarity, non-stationarity and early warning signals in economic networks. *Journal of Complex Networks* 3, 1-21 (2015).
25. Park, J. & Newman, M. E. Statistical mechanics of networks. *Physical Review E* 70, 066117 (2004).
26. Garlaschelli, D. & Loffredo, M. I. Multispecies grand-canonical models for networks with reciprocity. *Physical Review E* 73, 015101 (2006).
27. Robins, G., Pattison, P., Kalish, Y. & Lusher, D. An introduction to exponential random graph (p^*) models for social networks. *Social networks* 29, 173-191 (2007).
28. Engel, J., Pagano, A. & Scherer, M. Reconstructing the topology of financial

networks from degree distributions and reciprocity. *Journal of Multivariate Analysis* 172, 210-222 (2019).

29. Brailly, J., Favre, G., Chatellet, J. & Lazega, E. Embeddedness as a multilevel problem: A case study in economic sociology. *Social Networks* 44, 319-333 (2016).

30. Mele, A. A structural model of dense network formation. *Econometrica* 85, 825-850 (2017).

Page 10. Interpreting the network structure using ERGM

The network analysis shows that the guarantee network experienced dramatic changes during financial crisis and the stimulus program. The burst of mutual guarantees could be the main factor that influences the changes of the guarantee network. In this subsection, we used exponential random graph models (ERGM)²⁴ to verify the significance of the formation of mutual guarantee relationships. ERGM are a family of probabilistic models that identify the distribution over a specified set of graphs that maximizes the entropy subject to the known constraints^{25,26}. ERGM can effectively predict the edges as a function of these constraints, which usually represent the most important topological properties of the network structure²⁷. It has been applied to analyze various networks in finance and economy, such as^{28 29 30 23}.

Among all topological properties..... We use DCM as the null model, and find that the reciprocal edge (representing the mutual guarantee relationship) is statistically significant ($z\text{-score} > 3,000$) throughout the entire time period. On the other hand, the edge (representing both mutual and non-mutual guarantee relationship) is not significant. In addition, β_1 was negative and decreasing but β_2 was positive and kept increasing after the initiation of the stimulus program until March 2012, except a drop at the end of the stimulus program. These results indicate that firms were becoming increasingly likely to form mutual guarantee relationship. The ERGM analysis further demonstrates the significant role of mutual guarantee in the formation of the guarantee network.

Page 18-20. Exponential random graph models

Given the structure of real network, first we need to choose a set of topological properties as constraints, denoted as C_i^* ($i = 1, 2, 3 \dots k$), where k is the total number of constraints²³. We consider a set of networks \mathcal{G} , an ensemble of all networks of n nodes without self-loops, whose expected value of constraints $\langle C_i \rangle$ over \mathcal{G} is equal to that of the real guarantee network (C_i^*). The probability

of a network $G \in \mathcal{G}$ is denoted as $P(G)$. It has been proved that we can obtain the value of $P(G)$ under constraints C_i^* by maximizing the Gibbs entropy S , which is defined as follows,

$$S = - \sum_{G \in \mathcal{G}} P(G) \ln P(G), \quad (3)$$

with the constraints

$$\langle C_i \rangle = \sum_{G \in \mathcal{G}} P(G) C_i(G) = C_i^*, \quad (4)$$

and the normalization condition

$$\sum_{G \in \mathcal{G}} P(G) = 1, \quad (5)$$

where $C_i(G)$ is the value of C_i in network G .

Using the Lagrange multipliers, we can find that the maximum entropy is achieved for the distribution satisfying

$$\frac{\partial}{\partial P(G)} \left[S - \alpha \left(1 - \sum_{G \in \mathcal{G}} P(G) \right) - \sum_i \beta_i (\langle C_i \rangle - \sum_{G \in \mathcal{G}} P(G) C_i(G)) \right] = 0. \quad (6)$$

By solving equation (6), we obtain

$$-\ln P(G) - 1 + \alpha + \sum_i \beta_i C_i(G) = 0. \quad (7)$$

Or, equivalently, the probability of graph G is obtained as follows

$$P(G) = \frac{e^{H(G)}}{Z}, \quad (8)$$

with the graph Hamiltonian function

$$H(G) = \sum_i \beta_i C_i(G), \quad (9)$$

and the partition function

$$Z = e^{1-\alpha} = \sum_{G \in \mathcal{G}} e^{H(G)}. \quad (10)$$

We define the topological structure of a network G with an adjacency matrix \mathbf{A} , whose entries $a_{ij} = 1$ if node i and node j are connected, and $a_{ij} = 0$ otherwise.

In our study, we denote the real guarantee network by the particular matrix A^* .

In this study, we chose two constraints: edge number and reciprocal edge number of guarantee network. Please refer to Results section for discussions of why these two properties are the defining features of the Chinese guarantee network. We assume that the expected number of edges $\langle D(G) \rangle$ and the expected number of mutual edges $\langle R(G) \rangle$ of the real network are known. Then, the Hamiltonian of G is

$$H(G) = \beta_1 D(G) + \beta_2 R(G), \quad (11)$$

where $D(G) = \sum_{i<j} (a_{ij} + a_{ji})$ denotes the number of edges, and $R(G) = 2 \sum_{i<j} a_{ij} a_{ji}$ denotes the number of reciprocal edges. Given fixed number of nodes, $D(G)$ and $R(G)$ essentially evaluate the *density* and *reciprocity* of the network given the fixed number of nodes in the network, respectively.

Taking (11) into (10), the partition function for the complete system is formulated as

$$\begin{aligned} Z &= \sum_{\{a_{ij}\}} \exp \left(\sum_{i<j} [\beta_1 (a_{ij} + a_{ji}) + 2\beta_2 a_{ij} a_{ji}] \right) \\ &= \prod_{i<j} \sum_{a_{ij}=0,1} \sum_{a_{ji}=0,1} e^{\beta_1 (a_{ij} + a_{ji}) + 2\beta_2 a_{ij} a_{ji}} \\ &= \prod_{i<j} [1 + 2e^{\beta_1} + e^{2(\beta_1 + \beta_2)}] \\ &= [1 + 2e^{\beta_1} + e^{2(\beta_1 + \beta_2)}]^{\binom{n}{2}}. \end{aligned} \quad (12)$$

Then, the free energy of network is

$$F = \ln Z = \binom{n}{2} \ln(1 + 2e^{\beta_1} + e^{2(\beta_1 + \beta_2)}). \quad (13)$$

Finally, the expected values of edge and reciprocal edge are

$$\langle D(G) \rangle = \frac{\partial F}{\partial \beta_1} = n(n-1) \frac{e^{\beta_1} + e^{2(\beta_1 + \beta_2)}}{1 + 2e^{\beta_1} + e^{2(\beta_1 + \beta_2)}}, \quad (14)$$

$$\langle R(G) \rangle = \frac{\partial F}{\partial \beta_2} = n(n-1) \frac{e^{2(\beta_1 + \beta_2)}}{1 + 2e^{\beta_1} + e^{2(\beta_1 + \beta_2)}}. \quad (15)$$

We set the expected values of edge and mutual edge to be equal to the actual counts of edges and reciprocal edges, respectively. From equations (14) and (15), we can

obtain the values of β_1 and β_2 , which represents the log-odds of the probability of edges and the number of reciprocal edges, respectively. Then, the probability of edges and the probability of reciprocal edges are the inverse-logits of β_1 and β_2 , respectively. Thus, a large value of the coefficient indicates the high probability of forming the corresponding topology (edge and reciprocal edge). Note that the coefficient can be negative, indicating the low prevalence of such topology in the real network. For more details of interpretations of ERGM, please refer to ^{23 25}.

Supplementary Information. Page 3.

Supplementary Note 2: Significance Test with Directed Configuration Model

First, we use the Directed Configuration Model (DCM) to generate 10,000 random networks. The nodes in each generated network should have the same in- and out-degree as the real guarantee network.

Second, we calculate the number of sub-pattern i in the real and random networks.

Third, calculate the Z-score of sub-pattern i .

$$Z_i = \frac{N_i(G^*) - \langle N_i(G) \rangle}{\delta(N_i(G))},$$

where $N_i(G^*)$ is the occurrence of sub-pattern i in real network G^* , $\langle N_i(G) \rangle$ is the expected occurrence of 10000 DCM ensemble graph, and $\delta(N_i(G))$ is the standard deviation of a series of $N_i(G)$. Thus, the larger the Z_i -score, the more statistically significant the sub-pattern i .

Comment 1-3: *In what follows, examples of the issues that the authors should clarify are provided:*

- the techniques mentioned above are employed in a rather limited way. As an example, ERG models can be used to test the significance of a given set of observations (see also T. Squartini and D. Garlaschelli, J. Complex Netw. 3(1), 1-21 (2015)): in the present contribution, however, no significance analysis of the topological quantities of interest (i.e. the ones shown in figs. 2 and 3) has been carried out. It is only mentioned once that "A closer examination of the loan guarantee data revealed that isolated mutual guarantee relationship (2-node(%)) displays in significantly higher frequencies than would be expected for a random network (pvalue=0.0017)." but without

further specifying which random model has been tested and the details of the procedure that has been implemented to calculate the aforementioned statistical significance;

Response 1-3: Thank you very much for your constructive comments and detailed suggestions.

We agree with the reviewer that significance analysis of the topological quantities is essential. We followed the reviewer's suggestion to perform the significance test on the isolated mutual guarantee relationship (2-node (%)) with the DCM model specified as the null model being tested (Squartini & Garlaschelli 2015). Please refer to the detailed revisions copied below:

Page 4-5. Static topological and financial properties

A closer examination of the loan guarantee data reveals that isolated mutual guarantee relationship (2-node (%)) is significantly more frequent than that would be expected for a random network generated ($p\text{-value}=0.002$). Here, the random network ensemble is generated by the commonly adopted Directed Configuration Model (DCM), which is configured by the in-degree and out-degree sequences of the network. Please refer to (Squartini & Garlaschelli 2015)²³ for details.

Supplementary Information. Page 3.

Supplementary Note 2: Significance Test with Directed Configuration Model

.....

(Please refer to Response 1-2 for copied part.)

Comment 1-4: *The authors claim that "The empirical results of the ERGM model showed that both the density and reciprocity were significant ($p\text{-value}<0.001$) throughout the entire time period, as shown in Figure 5." What the authors mean is unclear since, by definition, the empirical density and reciprocity values are reproduced by the model: hence, their value cannot be significant in the usual, statistical sense.*

Response 1-4: We really appreciate the comment. We agree with the reviewer that we should clarify the significance here. In the revised manuscript, we adopted the suggested approach to analytically derive the values of the coefficients. DCM was selected as the null model to test the significance of the reciprocal edges. We copied the revised part below:

Supplementary Information. Page 3.

Supplementary Note 2: Significance Test with Directed Configuration Model

.....

(Please refer to Response 1-2 for copied part.)

Page 10. Interpreting the network structure using ERGM

Among all topological properties, network density and reciprocity are the defining characteristics of the guarantee network, and other properties are with similar patterns driven by the values of these two properties. Network density is the most basic property that had dramatic changes: first increasing because of many bankruptcies, and then decreasing because of many new loans and guarantees. Reciprocity is important because the stimulus program encouraged small firms to obtain loans by providing mutual guarantees to each other. Therefore, we consider the number of edges and the number of reciprocal edges in the ERGM. The probability function of G is as follows:

$$P(G) = \frac{e^{\beta_1 D(G) + \beta_2 R(G)}}{Z}, \quad (1)$$

where $D(G)$ denotes the number of edges, and $R(G)$ denotes the number of reciprocal edges. Z is the partition function defined by equation (10). Following the procedure in Methods section, the coefficient of $D(G)$ and $R(G)$ are obtained (Fig. 4). We use DCM as the null model, and find that the reciprocal edge (representing the mutual guarantee relationship) is statistically significant (z -score > 3,000) throughout the entire time period. On the other hand, the edge (representing both mutual and non-mutual guarantee relationship) is not significant. In addition, β_1 was negative and decreasing but β_2 was positive and kept increasing after the initiation of the stimulus program until March 2012, except a drop at the end of the stimulus program. These results indicate that firms were becoming increasingly likely to form mutual guarantee relationship. The ERGM analysis further demonstrates the significant role of mutual guarantee in the formation of the guarantee network.

Comment 1-5: *The authors say that "[...] the average clustering coefficient (C (%)) is also relatively large (0.97%~1.49%), indicating a strong "a friend's friend is also a friend" effect throughout the guarantee network, implying guarantee is a strong trust relationship.": "strong" with respect to what? Has a random model been chosen as a benchmark or other real-world networks are being considered for comparison?*

Response 1-5: Thank you very much for your comments. Here, we intended to indicate that the average clustering coefficient of the Chinese guarantee network is much higher than that for most real-world complex networks. In the revised manuscript, we followed the reviewer's suggestion to adopt the DCM as the null model to test the significance. The results showed that it is highly significant with $p\text{-value} < 0.001$. We copied the revised parts below:

Page 5. Static topological and financial properties

In addition, the average clustering coefficient (C (%)) is also relatively large (0.97%~1.49%), indicating a strong "a friend's friend is also a friend" transitivity effect throughout the guarantee network ($p\text{-value} < 0.001$ as compared to the DCM null model²³). The value is also much higher than most of the real-world complex networks¹². Such transitivity effect shows that the guarantee is a strong trust relationship.

Supplementary Information. Page 3.

Supplementary Note 2: Significance Test with Directed Configuration Model

.....

(Please refer to Response 1-2 for copied part.)

Comment 1-6: *The authors just consider the weakly connected component of the network under analysis; such a network, however, is directed: what about the behavior of the strongly connected component?*

Response 1-6: Thank you for pointing this out! We agree with the reviewer that we should also analyze the strongly connected component (SCC). In the revised manuscript, we have added both the static and dynamic analyses of the strongly connected component, including the size of largest SCC, number of SCC, and shortest path

length within the largest SCC. In general, the patterns are similar to those of WCC. Please refer to the updated Supplementary Table 1, Figures. 1e, 1f and 2h. and related discussions in the Section 2 (Results) copied below:

Page 5. Static topological and financial properties

Second, the largest weakly connected component (*WCC*) and the largest strongly connected component (*SCC*) stand for only 27.99% and 0.62% of all nodes in the network, respectively, indicating that the network is more decentralized than most other complex networks such as social networks and biological networks. Within the largest *WCC* and *SCC*, we observe the small-world effect as indicated by the small average shortest path length (10.10~10.54). In addition, the average clustering coefficient (*C* (%)) is also relatively large (0.97%~1.49%), indicating a strong “a friend’s friend is also a friend” transitivity effect throughout the guarantee network (p-value<0.001 as compared to the DCM null model²³). The value is also much higher than most of the real-world complex networks¹². Such transitivity effect shows that the guarantee is a strong trust relationship.

Supplementary Information 6. Supplementary Table 1: Static topological and financial properties

Measure	Phase 1 (04/07-11/08)		Phase 2 (12/08-12/10)		Phase 3 (01/11-03/12)		Whole period (01/07-03/12)	
	Mean	SD	Mean	SD	Mean	SD	Mean	SD
N	38343.62	818.265	48062.35	6425.67	67130.87	4167.02	48975.6	12323.52
E	36891.925	739.893	46981.12	6372.16	65825.87	4202.5	47769.52	12329.30
$\langle d \rangle$	0.96	0.0019132	0.98	0.01	0.98	0.01	0.97	0.01
λ_{in}	3.30	0.145	3.23	0.07	3.23	0.04	3.29	0.18
λ_{out}	2.30	0.03	2.74	0.15	2.76	0.05	2.58	0.25
Δ	2.51E-05	6.10E-07	2.07E-05	2.80E-06	1.47E-05	8.92E-07	2.1 E-05	4.7 E-06
C (%)	0.97	0.04	1.49	0.152	1.41	0.05	1.23	0.257
WCC	10583.955	180.67	13279.54	1726.90	18994.6	1656.6	13720.36	3560.46
WCC (%)	27.57	0.45	27.66	0.60	28.35	0.71	27.99	0.76
WCN	12301.575	155.62	15286.62	1897.23	21372.87	1698.35	15669.06	3871.35
SCC	211.15	35.95	325.32	33.45	446.73	29.98	307.69	103.65
SCC (%)	0.56	0.08	0.67	0.06	0.66	0.02	0.62	0.11
SCN	34850.05	1030.70	44392.24	5554.45	61542.2	3889.55	44861.80	11103.26

r (%)	13.84	0.19	14.80	0.41	14.36	0.35	14.37	0.51
2-node (%)	3.56	0.10	3.82	0.17	3.56	0.25	3.65	0.22
3-node (%)	1.15	0.09	1.34	0.11	1.34	0.05	1.26	0.14
l (weakly)	14.52	0.285	18.07	1.86	18.40	0.42	16.93	2.13
l (strongly)	10.54	1.75	10.10	0.78	10.24	0.48	10.12	1.33
AA (million)	109986.975	104.18	2446.23	60.65	2597.22	61.85	2387.25	194.79
AL (million)	20279.09	10.03	452.61	12.61	407.71	9.94	423.40	28.48
ALR (%)	60	1	61	8	62	4	61	1
ARL (%)	4.67	0.08	3.73	0.53	2.74	0.08	3.95	0.09

N: number of nodes; E: number of edges; $\langle d \rangle$: average out/in-degree; λ_{in} : power-law index of in-degree distribution; λ_{out} : power-law index of out-degree distribution; Δ : density; C: average clustering coefficient (directed); WCC: size of the largest weakly connected component (giant component); WCC (%): ratio of weakly connected giant component (%); WCN: weakly connected component number; SCC: size of the largest strongly connected component (giant component); SCC (%): ratio of strongly connected giant component (%); SCN: strongly connected component number; r: reciprocity; 2-node (%): ratio of isolated 2-node reciprocal component (%); 3-node(%): ratio of fully connected three nodes (%); l (weakly): average shortest path length in weakly connected giant component; l (strongly): average shortest path length in strongly connected giant component; AA (million): average assets of firms ; AL (million): average loans of firms; ALR (%): average leverage ratio (total liabilities/total assets) of firms. ARL (%): average ratio of listed firms (number of listed firm / number of all firms). Mean and SD are the mean value and standard deviation of different properties within Phase i (i=1,2,3). Phase 1: April 2007 to November 2008; Phase 2: December 2008 to December 2010; Phase 3: January 2011 to March 2012; whole period is from January 2007 to March 2012.

(Figure 1 is in next page.)

Page 7. Dynamic network analysis

Fig. 1. Evolution of the scale of the guarantee network. (a) $N (10^4)$: number of nodes. (b) $E (10^4)$: number of edges. (c) $WCC (10^4)$: size of the largest weakly connected component (giant component); $WCN (10^4)$: weakly connected component number. (d) $WCC (%)$: ratio of weakly connected giant component. (e) SCC : the size of the largest strongly connected component (giant component); $SCN (10^2)$: count of strongly connected component. (f) $SCC (%)$: ratio of strongly connected giant component. In the legend, bankruptcy of NCFC: bankruptcy of New Century Financial Corporation in April 2007; bankruptcy of LB: bankruptcy of Lehman Brothers in September 2008; start of the CESP: implementation of the Chinese economic stimulus program in December 2008; end of the CESP: end of the Chinese economic stimulus program in December 2010.

(Figure 2 is in next page.)

Fig. 2. Dynamics of the topological properties of the guarantee network. (a) $\Delta (10^{-5})$: density. (b) $\langle d \rangle$: average out/in-degree. (c) λ_{in} : power-law index of in-degree distribution. (d) λ_{out} : power-law index of out-degree distribution. (e) r (%): reciprocity. (f) 3-node (%): ratio of fully connected three nodes. (g) C : average clustering coefficient (directed). (h) l : average shortest path length in the weakly connected giant component and strongly connected giant component.

Comment 1-7: *The parameter k in equation 3 is set empirically to 1.5: why? Choices like these should be properly motivated.*

Response 1-7: Thank you very much for your comments. We agree with the reviewer that more clarifications should be provided for how to obtain the value of parameter k . In the revised manuscript, instead of empirically setting a value based on domain expertise, we adopted a regression-based approach, which learns the value of k by fitting a logistic regression as defined by equation (2) with real-world guarantee data. Please refer to the detailed revisions copied below:

$$P_i(t) = \frac{1}{1 + e^{-k\left(\frac{L_i(t) + \sum w_j(t)G_{ij}(t)}{A_i(t)} - \delta\right)}} \quad (2)$$

Page 12. Simulation analysis of the resilience of guarantee network

$k = 1.18$, which is obtained by learning from the real data. Specifically, we adopt equation (2) as a logistic regression to fit the real values of $P_i(t)$. Specifically, if firm i at time t defaulted, $P_i(t) = 1$, otherwise $P_i(t) = 0$. The values of other factors, including $A_i(t)$, $L_i(t)$, $G_{ij}(t)$, and δ of firm i , at time t were provided by our dataset. $p \in (0,1)$ represents the initial default ratio.

Reviewer 2's comments and our responses:

Comment 2-1: *In this manuscript, based complex network, the authors analyze the structure and evolution of the Chinese guarantee network with five years' worth of real-world data from January 2007 to March 2012. The authors characterize the global topological properties of the guarantee network, and find that the global financial crisis during 2007-2008 and economic policies in the aftermath (i.e., Chinese economic stimulus program and latter adjustments) had significant influence on the evolution of guarantee network structure.*

The main innovations are: the authors study on the risk of a comprehensive nationwide guarantee network and the influence of economic situation and national economic policies on the topological structure of guarantee network. Taking the 2007-2008 financial crisis and the subsequent Chinese economic stimulus program as two perfect natural experiments to investigate the influence of economic situation and national economic policies on the structure of guarantee network, as well as the associated contagion risk in the system.

This manuscript is well written and the idea is interesting.

Response 2-1: Thank you very much for the positive comments and encouragement! We believe that the empirical and simulation analyses of this comprehensive nationwide guarantee network during two natural experiments can provide valuable insights into the influence of economic situation and national economic policies on the structure of guarantee network, as well as the associated contagion risk in the system. This is of particular importance as we have been witnessing crisis more frequently recently, such as the social unrest and the ongoing novel coronavirus pandemic. It is important and timely to have such “big data” insights to facilitate effective responses to these extreme events.

Comment 2-2: *But I still have some additional comments. The authors explained the research motivation of this manuscript mainly from a data-driven perspective. I suggest that the authors give more analysis on the research problems.*

Response 2-2: Thank you very much for your suggestions. We followed the reviewer's suggestion to revise Section 1 (Introduction) with extensive discussions and justifications to motivate the research problems that we aim to address. The revised part is copied below:

Page 2-4. Introduction

A financial network represents a collection of entities (e.g. firms and banks), linked by mutually beneficial business relationships ^{1,2}.....

China has the biggest bank loan in the world. The credit market is the core of China's financial system, and the corresponding guarantee network might arguably be the largest credit networks in the world. With the outbreak of 2007-2008 global financial crisis,

Decision makers also recognized such risk and took actions to regulate the debt behavior..... A good understanding of the guarantee network's dynamic structure under pressure could enhance the economic policy-making through identifying the potential systemic risks such as the domino-like cascading failures. Facing the onset of financial crisis, it is of particular importance to examine the tradeoff between saving firms with government bailout and maintaining the resilience of the financial system. There is a critical need for data-driven quantitative studies of the guarantee network and its associated risk. However, how to leverage real-world guarantee data to model such behaviors and the overall structure of the guarantee network remains a challenge.

Network science presents a natural way to address the challenge in modeling guarantee networks.....

Existing research on guarantee networks mainly focused on the analytics of small sampled data with only dozens or hundreds of firms.....

By harnessing a comprehensive data provided by one of China's major regulatory bodies ranging 01/2007-03/2012, we investigate the structure and evolution of Chinese guarantee network to answer the following questions:

- What are the unique topological properties of the Chinese guarantee network?
- What is the influence of 2007-2008 global financial crisis, China's stimulus program and the following monetary policy adjustment on the topological structure of Chinese guarantee network?
- Does the change of topological structure of Chinese guarantee network influence the resilience of the system?

To the best of our knowledge, this is the first attempt to quantitatively characterize the evolution of the nationwide guarantee network. The empirical and simulation studies indicate that the financial crisis made the network more resilient, and conversely, the government bailout degenerated network resilience.

(Please refer to the manuscript for the fully revised Introduction section.)

Comment 2-3: *To analyze the structure and dynamics of Chinese guarantee system, the authors constructed dynamic guarantee networks using the guarantee relationships between firms. Each node represents a firm, and each edge connecting two nodes represents the existence of guarantee relationship between the two corresponding firms in a specific month. Does the author distinguish the size of the guarantee amount? As we know, large guarantee amounts should have a greater impact.*

Response 2-3: Thank you very much for your suggestions. We agree with the reviewer that the size of the guarantee amount matters in the guarantee network. Following the reviewer's suggestion, we incorporated the size of guarantee amount to construct a new directed weighted guarantee network, in which the weight on an edge represents the loan guarantee amount between two corresponding firms. We analyzed the weighted guarantee network and found that the results were consistent with the main unweighted analysis. Therefore, we included the analysis of the weighted guarantee network as the robustness analysis. Due to the length limit, we include the robustness analysis in the Supplementary Information. The revised part is copied below:

Supplementary Information Page 4.

Supplementary Note 3: Analysis of Weighted Guarantee Network

To check the robustness of the findings above, we constructed weighted guarantee networks with the amount of loans as the weight on edges, and did the same analyses. Here, the weight on each edge is set to be the amount of the loan guarantee, which indicates the trust between the two firms, as well as the risk associated with this guarantee relationship. Note that the values of the many topological properties (such as density and reciprocity) are not affected by the weight of edges. Supplementary Figure 1 presents the dynamic of four topological properties, network size, average in-/out-degree, average clustering coefficient and average shortest path length, which are re-calculated with the edge weight. In particular, the network size is the sum of all edge weights. The in-/out-degree of a node is the sum

of the edge weights for incoming/outcoming edges incident to the node. The clustering coefficient is defined as the geometric average of the weights of the subgraph edge²⁷. The average directed shortest path of weighted network is calculated in the strongly connected giant component of network with guarantee loan of firms as edge weight. Please refer to²⁸ for detailed definitions. We found that the patterns were consistent with the main results found by unweighted network. The general trend and change points were similar to their counterparts shown in Fig. 1 and 2 in the main manuscript.

Supplementary Fig. 1. Dynamics of the topological properties of the weighted guarantee network. (a) E : network size. (b) $\langle d \rangle$: average degree of weighted guarantee network. (c) C : average clustering coefficient of weighted guarantee network. (d) l : average directed shortest path length within strongly connected giant component of weighted guarantee network.

Comment 2-4: *The authors investigated the influence of 2007-2008 global financial crisis and China's stimulus program in the topological structure of Chinese guarantee network. What is the network topology of foreign companies or foreign banks in China from January 2007 to March 2012?*

Response 2-4: Thank you very much for the comments. Our dataset doesn't not contain foreign companies in China. It would be interesting future research to explore the positions and roles of foreign companies and banks in the dynamic guarantee network.

Comment 2-5: *Markov chain Monte Carlo (MCMC) method was adopted to estimate the parameters of exponential random graph model. Please give some details about the algorithm.*

Response 2-5: Thank you very much for your suggestions. In the revised manuscript, we have adopted an alternative approach (instead of MCMC) to analytically derive the values of coefficients in the ERGM. We describe the detailed derivations in the Method Section. The revised part is copied below:

Page 18-20. Exponential random graph models

Given the structure of real network, first we need to choose a set of topological properties as constraints, denoted as C_i^* ($i = 1, 2, 3 \dots k$), where k is the total number of constraints²³. We consider a set of networks \mathcal{G} , an ensemble of all networks of n nodes without self-loops, whose expected value of constraints $\langle C_i \rangle$ over \mathcal{G} is equal to that of the real guarantee network (C_i^*). The probability of a network $G \in \mathcal{G}$ is denoted as $P(G)$. It has been proved that we can obtain the value of $P(G)$ under constraints C_i^* by maximizing the Gibbs entropy S , which is defined as follows,

$$S = - \sum_{G \in \mathcal{G}} P(G) \ln P(G), \quad (3)$$

with the constraints

$$\langle C_i \rangle = \sum_{G \in \mathcal{G}} P(G) C_i(G) = C_i^*, \quad (4)$$

and the normalization condition

$$\sum_{G \in \mathcal{G}} P(G) = 1, \quad (5)$$

where $C_i(G)$ is the value of C_i in network G .

Using the Lagrange multipliers, we can find that the maximum entropy is achieved for the distribution satisfying

$$\frac{\partial}{\partial P(G)} \left[S - \alpha \left(1 - \sum_{G \in \mathcal{G}} P(G) \right) - \sum_i \beta_i (\langle C_i \rangle - \sum_{G \in \mathcal{G}} P(G) C_i(G)) \right] = 0. \quad (6)$$

By solving equation (6), we obtain

$$-\ln P(G) - 1 + \alpha + \sum_i \beta_i C_i(G) = 0. \quad (7)$$

Or, equivalently, the probability of graph G is obtained as follows

$$P(G) = \frac{e^{H(G)}}{Z}, \quad (8)$$

with the graph Hamiltonian function

$$H(G) = \sum_i \beta_i C_i(G), \quad (9)$$

and the partition function

$$Z = e^{1-\alpha} = \sum_{G \in \mathcal{G}} e^{H(G)}. \quad (10)$$

We define the topological structure of a network G with an adjacency matrix \mathbf{A} , whose entries $a_{ij} = 1$ if node i and node j are connected, and $a_{ij} = 0$ otherwise. In our study, we denote the real guarantee network by the particular matrix \mathbf{A}^* .

In this study, we chose two constraints: edge number and reciprocal edge number of guarantee network. Please refer to Results section for discussions of why these two properties are the defining features of the Chinese guarantee network. We assume that the expected number of edges $\langle D(G) \rangle$ and the expected number of mutual edges $\langle R(G) \rangle$ of the real network are known. Then, the Hamiltonian of G is

$$H(G) = \beta_1 D(G) + \beta_2 R(G), \quad (11)$$

where $D(G) = \sum_{i < j} (a_{ij} + a_{ji})$ denotes the number of edges, and $R(G) =$

$2 \sum_{i<j} a_{ij} a_{ji}$ denotes the number of reciprocal edges. Given fixed number of nodes, $R(G)$ and $D(G)$ essentially evaluate the *density* and *reciprocity* of the network given the fixed number of nodes in the network, respectively.

Taking (11) into (10), the partition function for the complete system is formulated as

$$\begin{aligned}
Z &= \sum_{\{a_{ij}\}} \exp \left(\sum_{i<j} [\beta_1 (a_{ij} + a_{ji}) + 2\beta_2 a_{ij}a_{ji}] \right) \\
&= \prod_{i<j} \sum_{a_{ij}=0,1} \sum_{a_{ji}=0,1} e^{\beta_1(a_{ij}+a_{ji})+2\beta_2 a_{ij}a_{ji}} \\
&= \prod_{i<j} [1 + 2e^{\beta_1} + e^{2(\beta_1+\beta_2)}] \\
&= [1 + 2e^{\beta_1} + e^{2(\beta_1+\beta_2)}]^{\binom{n}{2}}.
\end{aligned} \tag{12}$$

Then, the free energy of network is

$$F = \ln Z = \binom{n}{2} \ln(1 + 2e^{\beta_1} + e^{2(\beta_1+\beta_2)}). \tag{13}$$

Finally, the expected values of edge and reciprocal edge are

$$\langle D(G) \rangle = \frac{\partial F}{\partial \beta_1} = n(n-1) \frac{e^{\beta_1} + e^{2(\beta_1+\beta_2)}}{1 + 2e^{\beta_1} + e^{2(\beta_1+\beta_2)}}, \tag{14}$$

$$\langle R(G) \rangle = \frac{\partial F}{\partial \beta_2} = n(n-1) \frac{e^{2(\beta_1+\beta_2)}}{1 + 2e^{\beta_1} + e^{2(\beta_1+\beta_2)}}. \tag{15}$$

We set the expected values of edge and mutual edge to be equal to the actual counts of edges and reciprocal edges, respectively. From equations (14) and (15), we can obtain the values of β_1 and β_2 , which represents the log-odds of the probability of edges and the number of reciprocal edges, respectively. Then, the probability of edges and the probability of reciprocal edges are the inverse-logits of β_1 and β_2 , respectively. Thus, a large value of the coefficient indicates the high probability of

forming the corresponding topology (edge and reciprocal edge). Note that the coefficient can be negative, indicating the low prevalence of such topology in the real network. For more details of interpretations of ERGM, please refer to ^{23 25}.

Comment 2-6: *I suggest the author give corresponding implications from the trade-off between risk and return. If the next major economic and financial crisis breaks out, should the Chinese government adopt a grand economic stimulus plan?*

Response 2-6: Thank you very much for the comment! This is a very constructive suggestion. Following the reviewer's suggestion, we have significantly extended Section 4 (Conclusions) to discuss the implications of this research. Particularly, we discussed the generalizability of this China-based study, and the dilemma that the government bailout avoided the instantaneous catastrophic loss at the expense of making the financial network fragile. We suggest that the risk management need a consecutive endeavour that considers such trade-offs in a proactive manner. The methods and results of this study provide such benchmark data, simulation platform, and baseline methods to facilitate decision makers in developing effective responses to crisis. If the next economic and financial crisis breaks out (which is very likely to happen soon, sadly due to the global novel coronavirus pandemic), the decision makers should consider the sacrifice of system stability brought by the grand economic stimulus plan. We suggest that the mutual guarantee relationship should be regulated to reduce the risk of systemic cascading failure in the future. Apparently, more research is needed to further investigate this important problems.

The newly added discussions are copied below:

Page 13-15 Discussion

This research presents the first attempt to quantitatively characterize the evolution of the entire Chinese guarantee system as a complex network.....

Our study contributes to the literature through proposing a complex network approach to analyzing the guarantee relationships.....

In practice, this study indicates that although the mutual guarantee relationship could help low-quality firms obtain more loans, and reduce the risk of small-scale

arbitrary risk events, it could be the cause of large-scale cascading failures/defaults for the whole system. This research suggests that decision makers (e.g. government, central bank, and other authorities) control the prevalence of mutual guarantee relationships through adjusting monetary policies.

Importantly, much work about resilience of financial systems has focused on the theoretical analysis due to the unavailability of confidential financial data. Therefore, the real conditions and evolution of financial systems have not yet been fully understood. Here, with the nation-wide and real-world Chinese loan guarantee data, we show that the empirical results are opposite to the intuitions. Specifically, to take the effect of survivorship bias into consideration, the guarantee network is much more stable during the period of financial crisis with the surviving and robust firms, however, the stability keeps on decreasing with the implementation of stimulus program, for absorbing much more firms which should have been bankrupted without this government bailout. Our empirical findings would facilitate the study of stability of real-world financial system and also enable better strategies to be developed for policy makers to perform financial bailout.

The results of this study also pose an interesting dilemma. Financial crisis has made the financial network healthier by obsoleting the weak firms, but with instantaneous huge economic consequence. The government bailout avoided the instantaneous catastrophic loss at the expense of making the financial network fragile. In other words, government bailout could mitigate the current crisis by sacrificing the financial stability in the future. The 2019-2020 novel coronavirus pandemic in fact tells a similar story. Risk management needs a consecutive endeavour that considers such trade-offs in a proactive manner. The methods and results of this study provide such benchmark data, simulation platform, and baseline methods to facilitate decision makers in developing effective responses to crisis.

Although this study relies on the data of the Chinese guarantee network, the results and implications are still very beneficial from three perspectives.....

Comment 2-7: Finally, evolution of the Guarantee Network can be applied in stock market returns problems. I suggest that the authors pay attention to the following paper:

1. Dai Z., Zhu H. (2019) Forecasting stock market returns by combining Sum-of-the-parts and ensemble empirical mode decomposition, *Applied Economics*, doi:10.1080/00036846.2019.1688244.

Response 2-7: Thank you very much for your suggestions. We agree with the reviewer that this set of network science methodologies have good potential to be applied in other financial research, such as predicting the stock market return. We added the discussions of this future research in the Conclusion section. We also cited a few references including the suggested (Dai & Zhu 2019). The newly added part is copied below.

Page 14. Discussion

The insights into the evolution of the guarantee network have great potential to be applied in addressing other finance and economic research, such as forecasting stock market returns³³, credit allocation⁷ and international trade²³.

The added reference is below:

33. Dai, Z. & Zhu, H. Forecasting stock market returns by combining sum-of-the-parts and ensemble empirical mode decomposition. *Applied Economics*, 1-15 (2019).

7. Cong, L. W., Gao, H., Ponticelli, J. & Yang, X. Credit allocation under economic stimulus: Evidence from China. *The Review of Financial Studies* 32, 3412-3460 (2019).

23. Squartini, T. & Garlaschelli, D. Stationarity, non-stationarity and early warning signals in economic networks. *Journal of Complex Networks* 3, 1-21 (2015).

Comment 2-8: Additionally, there are some typos in the manuscript, e.g.,

(1) At page 4, "In section 2," should be "In Section 2,". Please check all of this manuscript.

(2) At the bottom of page 6, "both $p_{in}(k)$ and $p_{out}(k)$ follows" should be " both $p_{in}(k)$ and $p_{out}(k)$ follow ".

(3) Need page numbers.

Response 2-8: Thank you very much for your careful review. We have corrected these typos, and thoroughly polished the writing of the whole paper again.

Again, the authors would like to express our sincere gratitude to the reviewers and editor's valuable comments and constructive suggestions. Thank you very much!

REVIEWERS' COMMENTS:

Reviewer #1 (Remarks to the Author):

The authors have satisfactorily answered all issues that have been raised. I have only minor comments:

- the authors have, now, correctly employed the ERG formalism but I suggest them to make an effort to better distinguish the two different ways they employ it (i.e. they, first, use the DCM to test the statistical significance of the observed reciprocity; then, they explicitly encode it as a constraint in the model with two constraints only). This is not immediately obvious, especially at page 8 where, after presenting the "two-constraints" model, the authors say

"We use DCM as the null model, and find that the reciprocal edge (representing the mutual guarantee relationship) is statistically significant ($z\text{-score} > 3,000$) throughout the entire time period."

seemingly suggesting that formula 1 describes the DCM;

- I suggest the authors to explicitly name the model described by formula 1 as "reciprocity model", according to the original reference where it was proposed for the first time, i.e. J. Park and M.E.J. Newman, Phys. Rev. E 70, 066117 (2004);

- at page 15, the derivation of the partition function is correct, but the "double sum" passage is not written correctly: links, in fact, are no longer independent and one cannot sum separately over a_{ij} and a_{ji} . I suggest the authors to have a look at the reference D. Garlaschelli and M. I. Loffredo, Phys. Rev. E 73, 015101(R) (2006).

In my opinion, the paper can be judged as suitable for publication, after the issues above have been addressed.

Reviewer #2 (Remarks to the Author):

I've looked through the current version of this manuscript. The authors have revised the manuscript according to my suggestions. The paper is interesting and fits with the objectives of Nature Communications. The results presented within the revised manuscript are important for risk management. I recommend its publication in Nature Communications.

Responses to Reviewers' Comments

We would like to thank the editor and reviewers for their detailed and constructive comments. We really appreciate the help and encouragement during the peer-review process. We have revised the manuscript according to their comments.

Reviewer 1's comments and our responses:

Comment 1-1: *The authors have satisfactorily answered all issues that have been raised. I have only minor comments:*

- the authors have, now, correctly employed the ERG formalism but I suggest them to make an effort to better distinguish the two different ways they employ it (i.e. they, first, use the DCM to test the statistical significance of the observed reciprocity; then, they explicitly encode it as a constraint in the model with two constraints only). This is not immediately obvious, especially at page 8 where, after presenting the "two-constraints" model, the authors say "We use DCM as the null model, and find that the reciprocal edge (representing the mutual guarantee relationship) is statistically significant ($z\text{-score} > 3,000$) throughout the entire time period." seemingly suggesting that formula 1 describes the DCM;

- I suggest the authors to explicitly name the model described by formula 1 as "reciprocity model", according to the original reference where it was proposed for the first time, i.e. J. Park and M.E.J. Newman, Phys. Rev. E 70, 066117 (2004);

Response 1-1: Thank you very much for your encouragement and very constructive comments!

We agree with the reviewer that we should better distinguish the two different ways we employ it. In the revised manuscript, we introduce the test with DCM before we present formula (1) to motivate the use of an ERGM considering the number of edges and the number of reciprocal edges. We followed the reviewer's suggestion to name the model as the "reciprocity model" and cited the corresponding reference.

Comment 1-2: - at page 15, the derivation of the partition function is correct, but the "double sum" passage is not written correctly: links, in fact, are no longer independent and one cannot sum separately over a_{ij} and a_{ji} . I suggest the authors to have a look at the reference D. Garlaschelli and M. I. Loffredo, *Phys. Rev. E* 73, 015101(R) (2006).

Response 1-2: Thank you very much for pointing it out! It was a typo in the first line of Equation (12). We have corrected this typo in the equation and carefully gone through all derivations again to ensure that there is no more typo.

Reviewer 2's comments and our responses:

Comment 1-1: *I've looked through the current version of this manuscript. The authors have revised the manuscript according to my suggestions. The paper is interesting and fits with the objectives of Nature Communications. The results presented within the revised manuscript are important for risk management. I recommend its publication in Nature Communications.*

Response 1-1: Thank you very much for your encouragement and positive comments!

Again, the authors would like to express our sincere gratitude to the reviewers and editor's valuable comments and suggestions, which have greatly helped us to improve the quality of the paper. Thank you very much!